



# Regional significance of historical trends and step changes in Australian streamflow

Gnanathikkam, E. Amirthanathan[1], Mohammed A. Bari[2], Fitsum Woldemeskel[1], Narendra K. Tuteja[3*], Paul M. Feikema[1]

[1]Bureau of Meteorology, Melbourne, Australia.
[2]Bureau of Meteorology, Perth, Australia.
[3]Bureau of Meteorology, Canberra, Australia.
*Current address WaterNSW, Sydney

*Correspondence to*: Mohammed Bari (mohammed.bari@bom.gov.au)

**Abstract.** The Hydrologic Reference Stations is a network of 467 high quality streamflow gauging stations across Australia, developed and maintained by Bureau of Meteorology, as part of ongoing responsibility under the Water Act 2007. Main objectives of the service are to observe and detect climate-driven changes in observed streamflow and to provide a quality controlled dataset for research. We investigate linear and step changes in

streamflow across Australia in data from all 467 streamflow gauging stations. Data from 30 to 69 years duration ending in February 2019 was examined. We analysed data in terms of water year totals and for the four seasons. The commencement of water year varies across the country – mainly from February-March in the south to September-October in the north. We summarised our findings for each of the 12 Drainage Divisions defined by Australian Geospatial Fabric (Geofabric), and continental Australia as a whole. We used statistical tests to detect

and analyse linear and step changes in seasonal and annual streamflow. Linear trends were detected by Mann-Kendall – Variance Correction Approach (MK3), Block Bootstrap Approach (MK3bs) and Long Term Persistence (Mk4) tests. The Nonparametric Pettitt test was used for step change detection and identification. Regional significance of these changes at the drainage division scale was analysed and synthesised using the Walker test. The Murray Darling Basin, with Australia's largest river system, showed statistically significant

decreasing trends for the region in annual total and all four seasons. Drainage Divisions in New South Wales, Victoria and Tasmania showed significant annual and seasonal decreasing trends. Similar results were found in south-west Western Australia, South Australia and north-east Queensland. There was no significant spatial pattern observed in Central and mid-west Australia, one possibility being the sparse density of streamflow stations and or length of data. Only the Timor Sea drainage division in northern Australia showed increasing

trends and step changes in annual and seasonal streamflow and were regionally significant. Most of the step changes occurred during 1970-99. In the south-eastern part of Australia, majority of the step changes occurred in the 1990s, before the onset of the millennium drought. Long term linear trends in observed streamflow and its regional significance are consistent with observed changes in climate experienced across Australia. Findings from this study will assist water managers for long term infrastructure planning and management of water

resources under climate variability and change across Australia.



## 1   Introduction

Australia is the driest inhabited continent on Earth, receiving only 450mm/yr of rainfall on average (CSIRO and BOM, 2020). Rainfall amounts vary significantly across the country, with approximately 70 percent of the landmass being arid or semi-arid receiving less than 350mm/yr. The distribution and amount of rainfall across the country has influenced patterns of human settlement for more than 60,000 years (Williams, 2013). The continent also has unique topographic and geologic features. The central region is mostly arid or semi-arid, south-east and south-west corners having temperate, and the north having tropical climate (Stern et al., 2000) . The east and south-east coastal regions have mountain ranges.  Rainfall is higher and more reliable in coastal regions, except mid-west coastal regions of Western Australia. Elevation is another factor that has an important influence on rainfall, with mountainous areas such as northeast Queensland, southeast Australia and western Tasmania receiving higher rainfall (Holper, 2011). Rainfall is highly variable in both space and time compared to other continents. Together with unique topographic and geological features and distribution of rainfall result in the greatest interannual variability in streamflow (Nicholls et al., 1997; Poff et al., 2006), floods and droughts.

The 'Millennium Drought' between 1997 and 2009 is described as the worst drought on record for southeast Australia (Van Dijk et al., 2013). During 2001-2009, southeast Australia suffered the driest period since 1900 – the longest uninterrupted series of years with below median rainfall (Bureau of Meteorology data; http://www.bom.gov.au/cgi-bin/climate/change/timeseries.cgi). Discharge of the River Murray System during this period were half the previous recorded minimum.  As a response to the widespread social, financial and environmental impacts of drought, the federal government passed the Water Act 2007 (https://www.legislation.gov.au/Details/C2017C00151) legislation through the parliament heralding the implementation of the water security plan for Australia. One of the important roles in realising the water security plan is enhancing our understanding of Australia's water resources. The Hydrologic Reference Station service (http://www.bom.gov.au/water/hrs/index.shtml) was developed to provide greater insight of climate-driven changes in streamflow across Australia. Similar services exist in Canada, United States, Europe (Bradford and Marsh, 2003; Brimley et al., 1999; Coxon et al., 2020; Falcone et al., 2010; Lins, 2012; Whitfield et al., 2012) and in other areas across the globe (Alfieri et al., 2020).  In Canada, spatial and temporal trends in streamflow and other associated hydroclimatic variables show increasing and decreasing patterns in northern and mid-latitude catchments, respectively (Bawden et al., 2015; O'Neil et al., 2017). In the continental United States, streamflow analyses from 1940 to 2009 for 967 gauging stations, show overall decreasing trends, including higher annual maxima, and lower annual minima (Rice et al., 2015). Similar results were also found by Asadieh, et al. (2016) over the 1971-2001 period in the United States and other continents across the world. When comparing trends in minimally altered or regulated catchments in the United States, Hodgkins, et al. (2020) found there were larger changes in median streamflow compared to changes in annual 1-day high or 7-day low flows. Shifts in flood peaks and streamflow timing were found to be consistent with changes in rainfall, in both natural and managed catchments (Ficklin et al., 2018). In Finland, trend analysis revealed no changes in mean annual flow overall, but some changes in seasonal distribution of streamflow (Korhonen and Kuusisto, 2010). In south-east Asia and Africa, analyses of observed streamflow mostly show downward trends. In the Senegal river basin analysis of annual streamflow showed decreasing trends (Diop et al., 2018), while dry season flows increased by 6% at the Black Volta basin in West Africa over the period 2000 to 2013 (Akpoti et



al., 2016). Decreasing trends were also found in streamflow extremes across China (Li et al., 2020), and in streamflow in Huaihe River Basin, China (Pan et al., 2018). Similar trends were also evident in annual streamflow in West Borneo, Indonesia (Herawati et al., 2015).

Global-scale investigation of trends in annual maximum streamflow reveal decreasing trends tend to occur in Asia, Australia, the Mediterranean, the western and north-eastern United States, and northern Brazil, increasing

trends appear mostly in central North America, southern Brazil and the northern part of western Europe (Do et al., 2017). However, it was not clear how these changes relate to changes in rainfall. Analyses of streamflow trends, covering more than 30,000 gauges across the world, Gudmundsson, et al. (2019) found that where trends are present in a region, direction of the trend is often consistent across all trend analyses indicators including high, average and low flows for that specific region. Analyses of floods and extreme streamflow events across

the world show little to suggest that increases in heavy rainfall events at higher temperatures result in similar increases in streamflow (Wasko and Sharma, 2017). However, shifts in timing of flood peaks and changes in streamflow timing were found to be consistent with rainfall change and in a similar direction (Wasko et al., 2020).

Australian streams showed the greatest influence by interannual variability in flow (Poff et al., 2006). Chiew

and McMahon (1993) examined annual streamflow series of 30 unregulated Australian catchments to detect trends or changes in the means. They concluded that any changes were directly related to interannual variability rather than any changes in climate. Analysis of trends in Australian flood data from 491 stations (Ishak et al., 2010) indicated that about 30% showed trends in annual maximum flood, with downward and upward trends in southern and northern part of Australia, respectively. Other studies have investigated trends in selected

streamflow components in particular regions – in the southwest of Western Australia (Durrant and Byleveld, 2009; Petrone et al., 2010) and in southeast Victoria (Tran and Ng, 2009). In Victoria and the Australian Alps, assessment and reconstruction of catchment variability and trends in streamflow showed a decline during the period 1977−2012 (Fiddes and Timbal, 2016). Similar spatial pattern in streamflow reduction also occurred during the Millennium Drought between 1997-2009. Johnson et al (2016) reviewed historical trends and

variability in flood events across Australia and concluded that the link between trends in flood events and rainfall cannot be made due to the influence of climate processes such as temperature and evapotranspiration over different spatial and temporal scales. Similar results were also found by Wasko and Nathan (2020) and Sharma et al. (2018) – that changes in rainfall and soil moisture did not always explain trends in flooding.

The studies above undertook trend analysis of Australian rivers with limited spatial or temporal coverage of

streamflow data. This study undertakes a systematic appraisal of changes and trends in observed streamflow records in a large number of catchments across the country which are largely unaffected by human influence. Zhang et al. (2016) undertook the first comprehensive study employing the Hydrologic Reference Stations and analysed data until 2014 from 222 stations across Australia. That study investigated many components of streamflow including annual total, high and low flows, seasonal totals and baseflow components were analysed

and presented in the website http://www.bom.gov.au/water/hrs. The above study considered only the streamflow variables that are random and did not consider the effect of the autocorrelation structure and long-term persistence on the trend of streamflow variables. The Hydrologic Reference Stations service is now updated and contains information for 467 gauging stations and streamflow data to February 2019. In this study, we focus on



linear trends and step changes in annual and seasonal streamflow across different drainage divisions of
Australia, and its regional significance. Investigation of the driving force of changes in rainfall patterns on these
resulting trends in streamflow is out of scope of this study. Results from this study will benefit managers and
researchers in sustainable water management and long-term planning in water allocation, agricultural planning,
and hydropower.

## 2   Selection of gauging stations and data quality

### 2.1   Station selection guidelines

Guidelines for selection of the Hydrologic Reference Stations (HRS) were described in detail by
Turner et al. (2012). These include a minimum of 30 years of continuous data with less than 5% of
missing data in unimpaired catchments. In the recent update of the service, the Bureau of Meteorology
implemented two additional criteria: (i) the percentage of flow volume included as infilled data and,
(ii) the percentage of flow volume above the maximum gauged discharge. The thresholds for these two
criteria were: (i) a maximum of 10% for flow volume of infilled data and, (ii) a restriction in
extrapolated data to a maximum of 25%. Details of the station selection guidelines are presented in the
website. All of the 467 gauging stations that met these guidelines and were included in the updated
service are included in this study.

### 2.2   Updated number of gauging stations

#### 2.2.1   Gauging stations in 2020 update

Of the existing 222 gauging stations in the network prior to the service (Zhang et al., 2016), 12 are now
decommissioned by the data providing agencies, including 5 from Northern Territory, 2 from Victoria, 2 from
Western Australia, 1 from New South Wales and 2 from South Australia. The revised and updated selection
guidelines were applied to the remaining 210 gauging stations, and 179 of these stations passed the new
guidelines. More information about rating curves of 43 existing stations are now available which suggests that
these stations do not pass selection criteria. Therefore these 43 stations were removed from HRS service in 2020
(Table 1).

#### 2.2.2   New stations included in the service

Australia's instrumental record is relatively sparse before 1940, and few locations have continuous
rainfall measurement before 1900. At present, there are approximately 4,800 streamflow gauging
stations across Australia.  Many of these stations previously had insufficient data in Bureau's system to
be considered for the HRS service. Over recent years, more data has become available, and a set of
780 stations identified by Zhang et al. (2013) were selected for further investigation. These 780
unregulated, unimpaired catchments are widely spread across Australia (Fig. 1), and have undergone
strict quality assurance and quality control, including quality code checking for daily streamflow





records (Zhang et al., 2013). The total number of streamflow gauging stations that met the new guidelines and are now included in the service is 467, which is an increase from 222 stations (Table 1).

### 2.2.3 Reference period, quality control and catchment description

The HRS was updated with streamflow data ending in February 2019. This month was chosen to capture the 2018 water year for all stations. Commencement of recording of streamflow data varies across the country. The longest record begins from 1950s and the shortest one from 1980s. Figure 1 shows the location of stations and duration of record, and includes the 12 decommissioned stations. Data recorded at stations prior to 1950 was excluded from analysis, as the missing data were more than 5% before this time in most of the cases. Stations

with longest records are generally those in the high-value water resource catchments and populated areas along the coastal regions. Upstream catchment area also varied across the country (from 4.5 to 23,2846 km$^2$) and in different hydroclimatic regions (Table 2). Most of the catchment area ranged from 50 km$^2$ to 10,000 km$^2$ (Table 2). The number of stations distributed across different drainage divisions vary substantially – Murray Darling division having the largest number of stations while the South-Western Plateau has no stations at all.

Drainage divisions are defined according to Australian Hydrological Geospatial Fabric (Geofabric) (Atkinson et al., 2008). In the previous update of the HRS service (Zhang et al., 2016), there were no stations in Pilbara-Gascoyne and North Western Plateau divisions. In the current version there are 10 and 2 stations, respectively (Fig. 1, Table 2).

The continental Australia has a wide range of climate zones as defined by Köppen Climate Classification (Stern

et al., 2000) – including tropical region in the north, temperate regions in the south, grassland and desert in the vast interior (Fig. 1). Water year is defined in accordance with Australian Water Information Dictionary (Bureau of Meteorology, 2021). It varies across the country - begins February-March in the south and September-October in the north (Table 2). Annual average rainfall for each of the divisions vary from 201 to 1,333 mm, respectively. Annual average PET is generally higher than annual average rainfall. Therefore, streamflow

generation process in most of the divisions are controlled by water-limited environments (Milly et al., 2005) except for the Tasmanian division (Table 2). Figure 2a shows the relationship of catchment area with record length, while Fig. 2b shows the catchment area and the distribution of percentage of catchments, the distribution is reasonably uniformly distributed in the range between 45 to 12,000 km$^2$ and slightly skewed outside this range. Longest period of record is 69 years for several catchments having area from 14 to 15,850 km$^2$.

A quality-assurance, quality control (QA/QC) process was applied to observed time-series of daily streamflow from each gauging station. This process identified and removed erroneous data values such as negative and extreme values. The process of detection and removal was automated and then checked manually. The GR4J model (Perrin et al., 2003) was adopted to infill any missing data in accordance with the selection guidelines detailed in Section 2.1. The mean and standard deviation of Nash-Sutcliff efficiency for all 467 catchments was

0.74 and 0.12 respectively. As part of this process, a simple error correction procedure was used to ensure that initial and final estimated flows matched adjacent observed values. This was done by linearly interpolating the start and end of the infilled period to the observed flows. Through this process, a continuous quality-checked daily streamflow time series was created for trend analyses.





Table 1 Total number of stations

| Station features | Selection guidelines | Numbers |
| --- | --- | --- |
| Existing stations decommissioned by February 2019 | Passed | 8 |
| Existing stations operational and data up to February 2019 | Passed | 171 |
| Existing stations operational and decommissioned | Failed | 43 |
| New stations operational and data up to February 2019 | Passed | 288 |
| Total HRS stations | Passed | 467 |

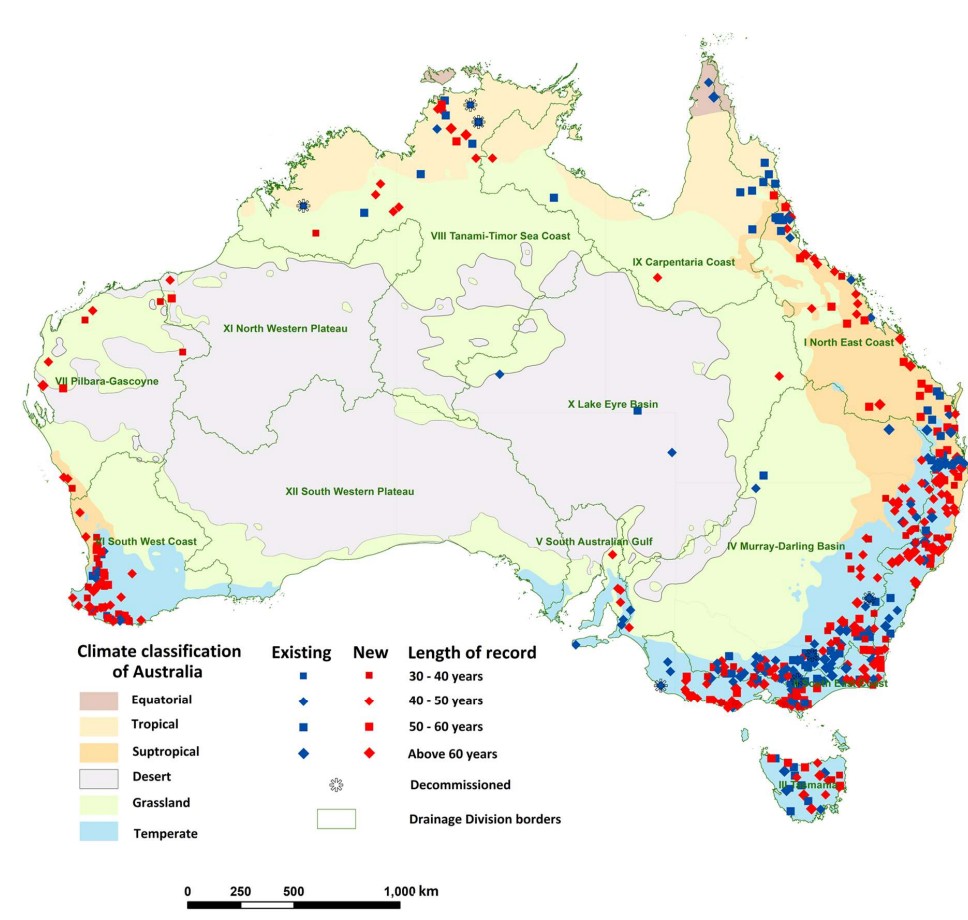





**Figure 1. Map of Australia showing climate zones, Drainage Divisions and location of all gauges, new, existing, decommissioned, all scaled to length of record.**

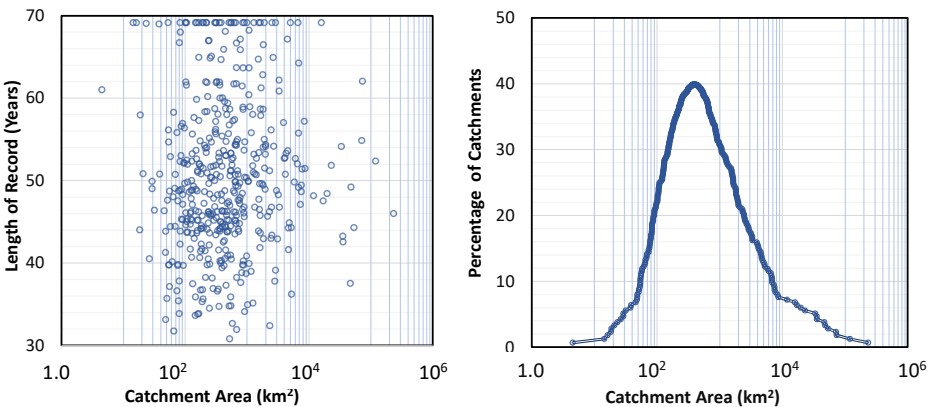

**Figure 2. Length of record, catchment area and distribution of percentage of catchments for HRS**

## 3    Methodology

Daily quality controlled and infilled daily data was accumulated to annual totals (based on water-year) and total for each of the four seasons. These five streamflow statistics formed the basis for trend analysis to capture annual and seasonal trends. Seasons are defined as winter (June to August), spring (September to November), summer (December to February), and Autumn (March to May).

There are generally two types of changes often observed in streamflow data sets – long term persistent trend (linear) and sudden abrupt (step) change. Linear trends generally occur due to long term changes in rainfall, temperature and/or evapotranspiration, while step changes generally happen resulting from sudden changes in flow generation thresholds. Statistical tests used to detect linear trends and step changes in streamflow generally fall into two categories – parametric and non-parametric tests.




Table 2: Metadata for Drainage Divisions and selected hydrologic reference stations

| Division number | Drainage division | Mean annual rainfall (mm) (1950-2018)* | Mean annual PET (mm) (1950-2018)* | Mean elevation (m ASL) | Number of stations | Water-year start | Smallest catchment (km²) | Largest catchment (km²) | Infilled volume (%) (Mean/SD) | Missing data (%) (Mean/SD) |
|---|---|---|---|---|---|---|---|---|---|---|
| I | North East Coast | 878 | 1855 | 173 | 66 | October | 16 | 35,326 | 0.7/1.7 | 0.5/1.0 |
| II | South East Coast | 885 | 1197 | 323 | 138 | March | 5 | 16,953 | 1.6/2.1 | 1.2/1.8 |
| III | Tasmania | 1333 | 853 | 199 | 25 | February | 18 | 3,285 | 0.7/1.1 | 0.8/1.3 |
| IV | Murray-Darling Basin | 484 | 1574 | 260 | 133 | March | 26 | 35,239 | 1.4/2.1 | 1.2/1.9 |
| V | South Australian Gulf | 313 | 1541 | 269 | 8 | February | 29 | 2,464 | 2.3/2.9 | 2.9/3.2 |
| VI | South West Coast | 439 | 1583 | 365 | 50 | March | 19 | 6,773 | 0.5/1.3 | 0.6/1.5 |
| VII | Pilbara-Gascoyne | 280 | 2098 | 162 | 10 | September | 907 | 72,902 | 0.1/0.2 | 0.1/0.2 |
| VIII | Tanami-Timor Sea Coast | 629 | 2152 | 339 | 21 | September | 65 | 47,652 | 5.2/2.8 | 6.0/6.0 |
| IX | Carpentaria Coast | 812 | 2143 | 293 | 10 | October | 333 | 8,638 | 4.1/2.9 | 5.5/3.2 |
| X | Lake Eyre Basin | 255 | 1849 | 312 | 4 | October | 3,324 | 232,846 | 0.1/0.2 | 0.9/0.9 |
| XI | North Western Plateau | 229 | 2115 | 359 | 2 | September | 6,503 | 53,323 | 5.6/5.2 | 4.3/3.2 |
| XII | South Western Plateau | 201 | 1773 | 297 | 0 | (No data) | (No data) | (No data) | (No data) | (No data) |

* Calculated from monthly gridded rainfall and PET data (5km by 5km) from AWAP (Raupach et al., 2009)





### 3.1 Linear trend analyses

We applied Theil Sen's approach (Sen, 1968; Theil, 1950), a non-parametric approach, to detect the magnitude of linear trend for each of the statistic (annual and four seasons total). In addition, trends were determined by using the non-parametric Mann–Kendall (MK) test (Kendall, 1975; Mann, 1945) because this technique is distribution-free, robust against outliers, and has a higher power for non-normally distributed data (Önöz and Bayazit, 2003; Yue et al., 2002) . It has also been commonly used in streamflow trend analyses (Abdul Aziz and Burn, 2006; Birsan et al., 2005; Dixon et al., 2006; Lins and Slack, 1999). The Mann–Kendall test requires input data to be serially uncorrelated. Any serial correlation in the data structure can leads to overestimation of the significance of trends (Hamed and Ramachandra Rao, 1998; von Storch, 1995; Yue et al., 2002). To overcome the effect of serial correlation of higher order (namely Short-Term Persistence (STP)), two techniques are used here. These include: (i) Modified Mann–Kendall test, commonly known as Variance Correction (MK3) method as proposed by Hamed and Rao (1998), and (ii) Block Bootstrap (MK3bs) method (Kundzewicz and Robson, 2000). The MK3 and MK3bs approaches are more suitable when the time series shows higher order serial dependencies. Apart from Short Term Persistence (STP), the presence of Long-Term Persistence (LTP), or Hurst phenomenon, has been identified as a major source of uncertainty when analysing hydroclimatic data series (Koutsoyiannis, 2003; Kumar et al., 2009). To incorporate LTP behaviour in the Mann–Kendall test, the technique proposed by Hamed (2008) is used in this study. Using these three modified versions of Mann–Kendall test allows comparison of differences and check the validity of results for Australia. Non-parametric Pettitt test (Pettitt, 1979) and distribution free CUSUM test (Chiew, and Siriwardena, 2005) are generally used to detect abrupt step changes in streamflow. The Pettit test was used to detect step change for all five streamflow statistics, as it performs better than others for detecting step change and identifying change point (Villarini et al., 2009). Finally, the regional significance of any linear trends and step changes was tested using the Walker test (Wilks, 2006).

The Theil-Sen Slope approach, original Mann–Kendall (MK) test, three modified versions of the Mann–Kendall test, Pettitt Test and Walker test are briefly described in this section. Trend detection analysis may lead to misleading results when serial correlation (STP) and long-term persistence (LTP) in the streamflow data are ignored. The modified versions of Mann-Kendall tests, MK3, MK3bs and MK4 that account for STP, STP and LTP respectively are used to identify trends in streamflow data. A more detailed description can be found in Mann, 1945; Kendall, 1975; Hamed and Rao, 1998; Koutsoyiannis, 2003; Hamed, 2008; Pettitt, 1979; Wilks, 2006; Kumar et al., 2009; Zamani et al., 2017; Su et al., 2018; Kundzewicz and Robson, 2000).

We used Theil-Sen estimator and three different forms of Mann Kendall tests for linear trend analyses: (i) variance correction approach, (ii) Block Bootstrap approach and (iii) Long Term persistence.

#### 3.1.1 Theil-Sen approach

The magnitude of trend is obtained using the Theil-Sen approach (Theil, 1950; Sen, 1968), where the magnitude of the slope of the trend is estimated as:

$$\beta = Median \ [(x_i - x_j)(i - j)]; \quad \text{for all} \ \ j<i \tag{1}$$



where $x_i$ and $x_j$ are streamflow data (annual and all seasons) at time points i and j, respectively. If the time series has n values, then there will be $N = n(n-1)/2$ slope estimates and Theil-Sen slope β is taken as the median of these N values.

### 3.1.2    Independent Mann-Kendall (MK1) test

The Mann-Kendall test statistic (S) for a series $x_1, x_2, x_3,…,x_n$ is given by

$$S = \sum_{i=1}^{n-1} \sum_{j=i+1}^{n} sgn(x_j - x_i), \tag{2}$$

where

$$sgn(\theta) = \begin{cases} 1 & if\ \theta > 0 \\ 0 & if\ \theta = 0 \\ -1 & if\ \theta < 0 \end{cases} \tag{3}$$

The test statistic S is approximately normally distributed for n ≥ 8 with zero mean.  Variance as given as

$$V(S) = [n(n-1)(2n+5) - \sum_{i=1}^{m}(t_i - 1)(2t_i + 5)t_i\ ]/18 \tag{4}$$

where $m$ is the number of tied groups and $t_i$ is the number of data in the i[th] tied group. The standardised test statistic Z (standard normal distribution) is given as

$$Z = \frac{(S-1)}{\sqrt{V(S)}}; \qquad if\ S > 0 \tag{5a}$$

$$Z = 0; \qquad if\ S = 0 \tag{5b}$$

$$Z = \frac{(S+1)}{\sqrt{V(S)}}; \qquad if\ S < 0 \tag{5c}$$

To identify and address the short term and long-term persistence in a streamflow series, we used three modified versions the of Mann Kendall test for linear trend analyses: (i) Variance Correction approach, (ii) Block Boot Strap approach and (iii) Long Term persistence approach.

### 3.1.3    Mann-Kendall (MK3) test – Variance Correction Approach

This modified Mann–Kendall test, proposed by Hamed and Rao (1998), considers all the significant autocorrelation structure in a time series. The series $x_i$ was ranked and autocorrelation coefficient of rank $i$ of time series, was obtained to consider only the significant terms at 10% significance level. Autocorrelation becomes insignificant after a lag of three (Rao et al., 2003).  The effect of all significant autocorrelation coefficients in the dataset was removed by using a modified variance of $S$, described as $V(S)^*$ by

$$V(S)^* = V(S) \frac{n}{n^*} \tag{6}$$

where $n^*$ represents the effective sample size. The $\frac{n}{n^*}$ ratio was computed directly from the equation proposed by Hamed and Rao (1998) as

$$\frac{n}{n^*} = 1 + \frac{2}{n(n-1)(n-2)} \sum_{i=1}^{n-1}(n-i)(n-i-1)(n-i-2)r_i \tag{7}$$

where $n$ is the number of observations; and $r_i$ is lag-i significant autocorrelation coefficient of rank $i$ of time 275    series.  $V(S)^*$ is calculated using Eq. (6) and using $V(S)$ from Eq. (4). Finally, the Mann-Kendall $Z$ was tested for significance of trend comparing it with threshold levels.





### 3.1.4    Mann-Kendall (MK3bs) test – Block Bootstrap (BBS) Approach

The Block Bootstrap (BBS) approach by Önöz and Bayazit (2012) was used to mitigate effects of serial correlation in datasets by performing bootstrapping (in blocks of data) so that the autocorrelation in the data is replicated. Data are resampled in blocks many times to estimate the significance of the observed Mann-Kendall test statistic S from the data sample while reflecting the serial correlation present in the dataset (Burn et al., 2016). Block length should be chosen so that data points in adjacent block are more or less independent. Khaliq et al. (2009) provide a detailed description of the steps involved in implementing the BBS approach.

In block bootstrapping, the simulation size $N_s$ (the number of bootstrap samples to be generated in each case) and the block length $L_b$ are the parameters whose values are to be chosen. The simulation size ($N_s$) is related to the level of significance required and the loss of power that can be allowed. Svensson et al. (2005) found that $N_s$= 2000 samples resulted in good stability in significance level estimates $N_s$Block length chosen depends on autocorrelation of the data, and should be larger than the *lag k* of the smallest significant autocorrelation coefficient $R_i$ Svensson et al. (2005) found that blocks of lengths of 5 were generally sufficient for most streamflow series.

Autocorrelation in time series varies across different sites, and Politis (2003) recommends the block length $L_b$ is based on individual time series. To identify optimal block length $L_b$ for individual annual and seasonal streamflow time series, an automatic block length selection procedure proposed by Politis and White (2004) was used. The optimal block length obtained for annual and for each of the four seasonal streamflow time series for 467 gauging stations are shown in Fig. 3 below. In general, more than a third of the stations exhibit significant serial correlation of 2 or more ($L_b \geq 3$) and about 15% are serially uncorrelated ( $L_b = 1$).

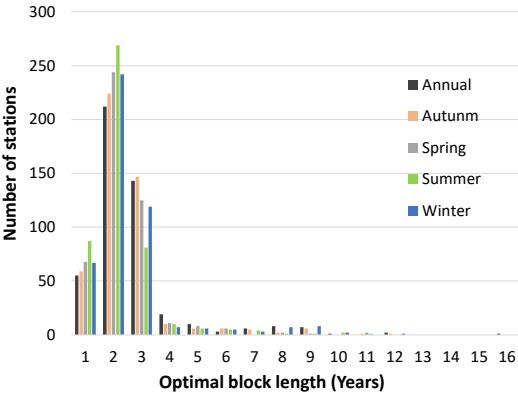

**Figure 3: Optimal block length for annual and four seasonal streamflow time series for all 467 stations**

We estimated the significance of the original Mann-Kendall test statistic $S$ (Eq. 2) of the observed data from the simulated distribution, developed from resampled distribution of S from the moving block bootstrap procedure. Only if the original test statistic lies within the tails of the simulated distribution, the test statistic is likely to be significant (Do et al., 2017; Khaliq et al., 2009).


### 3.1.5 Mann-Kendall (MK4) test – Long Term Persistence (LTP)

This version of the Mann-Kendall method is proposed by Hamed (2008) and was described by Kumar et al. (2009) and it considers the Hurst coefficient, H, (Hurst, 1951) of a series for Long-Term Persistence (LTP). The coefficient H is used as a measure of long-term memory, i.e., autocorrelation of the time series. A value of 0.5 for H indicates a true random walk, which implies the time-series has no memory for previous values of observations. A value of H between 0.5 (0) and 1 (0.5) indicates a time-series with positive (negative) autocorrelation [e.g. an increase (decrease) between observations will probably followed by another increase (decrease)].

In this study, the following steps were carried out to apply the MK4:

1.  Calculation of Hurst coefficient (H): A new trend free time series $x_i'$ is calculated using

$$x_i' = x_i - (\beta \times i) \tag{8}$$

where $\beta$ is the slope of a trend line using the Theil-Sen's approach and $x_i$ is the streamflow data. Using the ranks of trend-free series $[x_i': (i = 1:n)]$ designated by $R_i$, the standardised $Z_i$ variate (the equivalent normal variates of the ranks of de-trended time series) is computed as

$$Z_i = \varphi^{-1}\left(\frac{R_i}{n+1}\right) \tag{9}$$

where $n$ is the number of streamflow data; and $\varphi^{-1}$ is the inverse of the normal distribution function. Considering the hypothesis that the hydrometeorological processes exhibit scale invariant properties at any scale greater than annual (Koutsoyiannis, 2003), the elements of the Hurst matrix for a given H is computed as

$$C_n(H) = \left[\rho_{|j-i|}(H)\right] \qquad \text{for } i = 1:n, j = 1:n \tag{10}$$

In the above equation $\rho_l(H)$ is lag-$l$ autocorrelation coefficient for a given H and calculated by

$$\rho_l(H) = \frac{1}{2}\left(|l+1|^{2H} - 2l^{2H} + |l-1|^{2H}\right) \text{ for } l > 0 \tag{11}$$

The log-likelihood function of n Normal observations with a scaling coefficient H is given by Eq. (12) (McLeod and Hipel, 1978), where the accurate value of H can be computed by maximising the function of H as follows:

$$logL(H) = -\frac{1}{2}\log|C_n(H)| - \frac{Z^T|C_n(H)|^{-1}Z}{2\gamma_0} \tag{12}$$

In the above equation, $Z^T$ is the transpose of vector $Z$ obtained from Eq. (9); $\gamma_0$ equates the variance of $Z_i$ and $C_n(H)$ and $C_n(H)^{-1}$ are Hurst matrix and inverse of the Hurst matrix, respectively. These two last matrices can be obtained using Eq. (10). To maximise $LogL(H)$, H is assumed to be in the range of 0.5–0.98 and the mentioned function is computed for a given H. The procedure is repeated for other H values with 0.01 steps. The H value producing the maximum value of $LogL(H)$ is taken as the answer.

2.  Mean and standard deviation of H: According to Hamed (2008), the mean and standard deviation of H are expressed as a function of $n$ as follows:





$$\mu_H = 0.5 - 2.87n^{-0.9067} \tag{13}$$

$$\sigma_H = 0.77654n^{-0.5} - 0.0062 \tag{14}$$

Then $Z_{cal}$ is calculated as

$$z_{cal} = \frac{H - \mu_H}{\sigma_H} \tag{15}$$

This $Z_{cal}$, obtained from Eq. (15) was tested for significance of trend at the 10% significance level. The MK4 procedure was continued if $Z_{cal}$ was greater than the critical normal value (1.645), otherwise the procedure for independent Mann-Kendall (MK1) test was adapted as the LTP is not significant.

3. Significant H values (LTP is significant): The modified variance for the $S$ statistic was computed as recommended by Kumar et al. (2009) and Hamed (2009) as:

$$V(S)^{H'} = \sum_{i=1}^{n-1}\sum_{j=i+1}^{n}\sum_{k=1}^{n-1}\sum_{l=k+1}^{n} \frac{2}{\pi}\sin^{-1}\left(\frac{\rho|j-l| - \rho|i-l| - \rho|j-k| + \rho|i-k|}{\sqrt{(2-2\rho|i-j|)(2-2\rho|k-l|)}}\right) \tag{16}$$

In the above equation $\rho_l$ is calculated using Eq. (11) for a given value of H. Since $V(S)^{H'}$ is a biased estimator, we have corrected it for bias using

$$V(S)^H = V(S)^{H'} \times B \tag{17}$$

where $B$ is expressed as a function of sample size, $n$, as follows (Hamed 2008; Kumar et al. 2009):

$$B = a_0 + a_1H + a_2H^2 + a_3H^3 + a_4H^4 \tag{18}$$

$$a_0 = \frac{1.0024n - 2.5681}{n + 18.6693} \tag{19a}$$

$$a_1 = \frac{-2.2510n + 157.2075}{n + 9.2245} \tag{19b}$$

$$a_2 = \frac{15.3402n - 188.6140}{n + 5.8917} \tag{19c}$$

$$a_3 = \frac{-31.4258n + 549.8599}{n - 1.1040} \tag{19d}$$

$$a_4 = \frac{20.7988n - 419.0402}{n - 1.9248} \tag{19e}$$

When using the MK4 method, $V(S)^H$ obtained from Eq. (17) was used as $V(S)$ in Eq. (5) of the MK1. The significance of Z was then tested for significance of a trend.

### 3.2 Pettitt test for step change

We used the non-parametric Pettitt test (Pettitt, 1979) to detect step changes in annual and seasonal streamflow. This procedure is least sensitive to outliers, and skewed distributions of the streamflow datasets in comparison to other methods used for detecting step changes, makes it most suitable for this analysis (Sagarika et al., 2014). This test can identify anomalies in the mean and, or, median streamflow when the time of step change is unclear. It uses a version of the Mann–Whitney statistics to quantify the significance of probabilities by testing



two samples from the same population. In the Pettitt test the p-value is computed in a manner that adjusts for the fact that the method is designed to find the most advantageous point in the record to consider as the change point (Helsel et al., 2020; Villarini et al., 2009).

Using Pettitt (1979), let us assume $n$ to be the length of the time series and $\tau$ to be the year of the step change. Viewing the time series as two samples $x_1, \ldots, x_\tau$ and $x_{\tau+1}, \ldots, x_n$, an index, $V_\tau$ can be defined as:

$$V_\tau = \sum_{j=1}^{n} sgn(x_\tau - x_j) \tag{20}$$

where $sgn(x)$ is the same as $sgn(\theta)$ in Eq. (3) and $U_\tau$ is defined in Equation (21).

$$U_\tau = \sum_{i=0}^{\tau} V_i \tag{21}$$

When a significant step change exists in a time series, a graph between $|U_\tau|$ and $\tau$ increases up to the step change point and then decreases again, or vice versa. However, in the absence of a step change point, the graph would continually increase or decrease to the end of the time series. The most significant step change point $\tau$ is established at the point where $|U_\tau|$ is maximum, given as $K_n$ and defined by

$$K_n = \max_{1 \leq t \leq n} |U_\tau| \tag{22}$$

In a year where $|U_\tau|$ is the maximum, the significance probability associated with $K_n$ is approximated by

$$p = 2\, e^{\left(\frac{-6K_n^2}{n^3+n^2}\right)} \tag{23}$$

where the approximation holds and is accurate to two decimal places, for $p < 0.5$ (Pettitt, 1979).

In this study, we adopted a significance level of $p \leq 0.10$, and evaluated the direction of change. The minimum value of $U_\tau$, extracted by $K_n$ indicates positive change, and a maximum value indicates a negative change.

### 3.3 Test for regional significance

We assessed the regional significance of trends detected at a local point scale to examine if similar trends were also detected at neighbouring locations. The main objective is to assess whether the number of locations with significant trends occur at a regional scale or not. We applied Walker's test (Wilks, 2006; Sagarika et. al., 2014) in detecting regional significance of linear and step changes in streamflow time series – for annual total and across the four seasons at a 90% confidence level ($p \leq 0.10$) for all 467 stations across Australia.

The Walker's test considers a set of $K$ independent MK and Pettitt tests, all of whose null hypothesis are assumed to be true (i.e. corresponding p-values are assumed uniformly distributed as U(0, 1)). We further assume that $p_{(1)}$ is the smallest of the $p$ value set. In this case, the probability distribution $p_{(1)}$ is given by

$$f[p_{(1)}] = \frac{K!}{(1)!(K-1)!} p_{(1)}^0 [1 - p_{(1)}]^{K-1} \qquad 0 \leq p_{(1)} \leq 1 \tag{24}$$

$$f[p_{(1)}] = K[1 - p_{(1)}]^{K-1} \tag{25}$$



To reject the global null hypothesis that all $K$ local null hypotheses are true (i.e., to declare field significance), $p_1$ must be no larger than a critical value $p_{walker}$. The critical value for this global test can be obtained using

$$\propto_{global} = K \int_0^{p_{walker}} \left[1 - p_{(1)}\right]^{K-1} dp_{(1)} \tag{26a}$$

$$\propto_{globa} = 1 - (1 - p_{walker})^K \tag{26b}$$

$$p_{walker} = 1 - (1 - \propto_{global})^{1/K} \tag{27}$$

A global null hypothesis may be rejected at the $\propto_{global}$ (0.10) level if the smallest of $K$ independent local $p$ value is less than or equal to $p_{walker}$.

## 4    Results

For water year and the four seasons, linear trend considering short-term (STP) and long-term (LTP) persistence and the step changes were evaluated at a significance level of $p < 0.10$ for each streamflow station. The water year begins in September/October for the northern part of Australia (drainage divisions I, VII, VIII, IX, X) and in February/March for the southern part of the country (drainage divisions II, III, IV, V, VI) (Fig. 1, Table 1). We undertook trend analysis for annual and seasonal streamflows in these two regions of Australia separately.

### 4.1    Linear trends

#### 4.1.1    Theil-Sen approach

Box plots of trend slopes estimated by Theil-Sen's approach for annual and seasonal streamflow in southern and northern divisions that are significant are shown in Fig. 4. For example. the trend slope for a site in South Esk River in Tasmania is shown in Fig. 5a, where the decrease in annual streamflow is 5.91 GL/year (1.8 mm/year). For drainage divisions in the south, the medians of slopes for all annual and seasonal streamflows were less than zero. The lowest trend line slope (−2.07 mm/year) occurred in spring (Sep-Nov) and the median of slopes in autumn (Mar-May) is less negative than the other three seasons in the southern parts of Australia (Fig. 4a). Streamflow in all four seasons showed a downward trend in the southern divisions. Approximately 60% of annual streamflow volumes in the southern drainage divisions occur in winter and spring (between Jun-Nov). For annual flows in the south, slopes were between -6.5 to -0.07 mm/year (Fig. 4a) and the median of slopes was approximately -1.79 mm/year.

Medians of slopes for annual and seasonal streamflow were located close to, or above zero for drainage divisions in the northern part of Australia (Fig. 4b). The highest (+7.13 mm/year) trend line slope occurred in the summer (Dec-Feb). The median of trend line slopes in summer (+1.02 mm/year) is higher than that in the other three seasons (all <0.01 mm/year) in the northern divisions. Streamflow in all seasons, except in summer, showed negligible trends in northern parts of Australia. More than 90% of the annual streamflow volumes in the northern drainage divisions occur in summer and autumn (between Dec-May). For annual flows, slopes were between +12.1 and -6.0 mm/year (Fig. 4b) and the median of slopes was approximately +1.46 mm/year. Fig. 5a





shows a typical example of linear trend analysis using Theil Sen's approach for annual streamflow in southern

Australia.

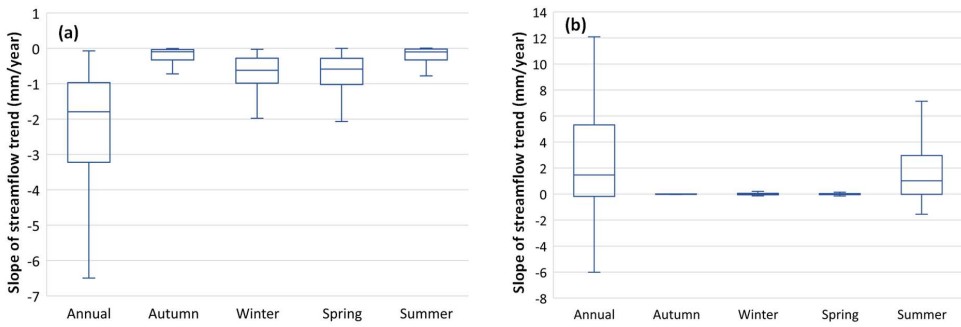

**Figure 4: Box plot of Theil-Sen's slope in mm/year for annual and seasonal streamflow for drainage
divisions in (a) Southern and (b) Northern Australia**

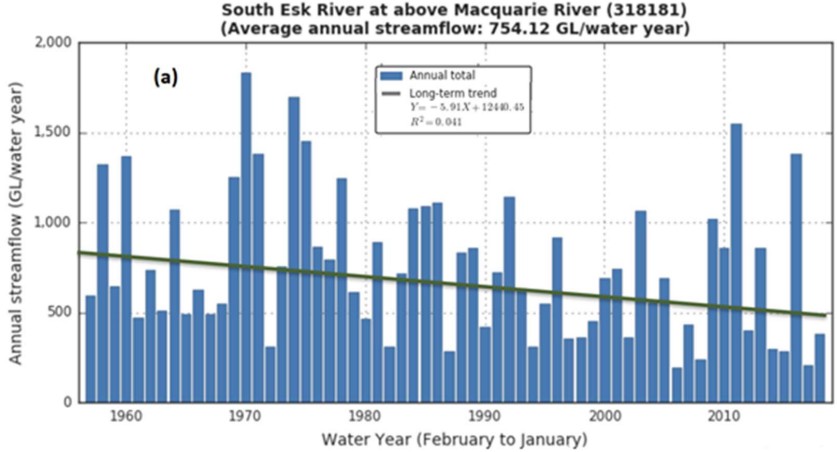

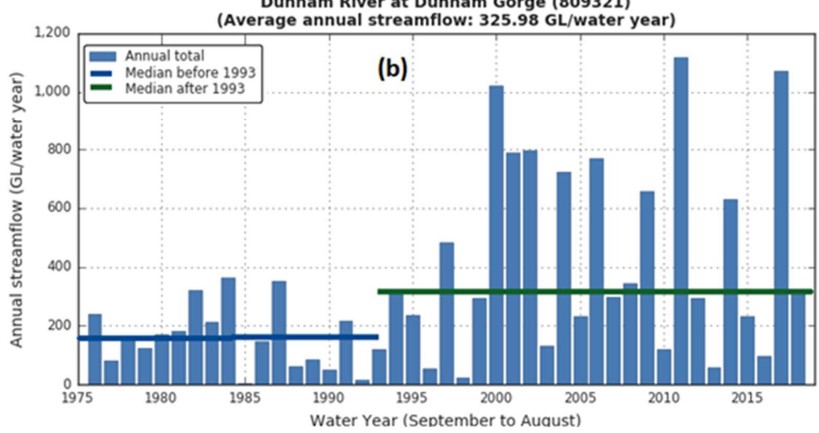

**Figure 5: Typical examples of (a) trend and (b) step change in annual streamflow in southern and
northern Australia**



### 4.1.2 Linear trend – the Short (STP) and Long (LTP) Term Persistence

Streamflow data from stations that are significantly autocorrelated (lag-1 or more) at $p < 0.10$ and H values that are significant at $p < 0.10$ are considered to have short-term (STP) and long-term (LTP) persistence, respectively. To analyse the presence of STP, results from the MK3 test were examined. In the South East Coast (II), South West Coast (VI) coasts and southern Murray-Darling Basin (IV), most stations showed significant STP or LTP, or both STP and LTP for the water–year and for all four seasons (Fig. 6). Across water years and all seasons, the percentage of stations with STP was greater than those with only LTP or both STP and LTP. For water years, data from 88% of stations showed STP and 28% of showed LTP across Australia. However, LTP was evident mainly in the southern and south-eastern parts of Australia (divisions II, III, IV, V, VI) and in the north in the Carpentaria Coast (IX). Seasonally, autumn had the highest number of stations (221) with LTP persistence.

Data from stations with significant correlation at $p < 0.10$ in STP and LTP were tested for trends (Table 3) using MK3 sand MK4 tests, respectively. Of the 412 stations with significant STP across water years, 196 stations showed trends from the MK3 test. Of 130 stations with data showing significant LTP, 14 had significant trends from the MK4 test. In drainage divisions II to VI in the south, nearly half of stations showed significant trends with STP. However, results vary slightly across different drainage divisions for the four seasons. In autumn, 33% of the 87% of stations with significant STP showed trends, while 6% of the 47% of stations with significant LTP showed trends. Similarly, in winter, 39% out of 86% of stations with STP, showed trends, while 3% of the 27% of stations with LTP, showed trends. In spring, 44% of the 85% of stations with significant STP, showed trends while 2% of the 24% of stations with significant LTP, showed trends. In summer, 36% of 81% of stations with significant STP, showed trends while 3% of 19% of stations showing LTP, showed trends.




Table 3 Summary of stations with short-term persistence (STP) and long-term persistence (LTP) of streamflow, and stations that showed trends under MK3 and MK4 tests across drainage divisions for water year and the four seasons at p ≤ 0.10.

| Drainage division | Drainage division name (number of stations) | Water year | | Autumn | | Winter | | Spring | | Summer | |
|---|---|---|---|---|---|---|---|---|---|---|---|
| | | STP/Trend | LTP/Trend | STP/Trend | LTP/Trend | STP/Trend | LTP/Trend | STP/Trend | LTP/Trend | STP/Trend | LTP/Trend |
| I | North East Coast (66) | 59/4 | 23/0 | 57/2 | 18/0 | 51/18 | 31/1 | 60/15 | 22/0 | 46/5 | 9/0 |
| II | South East Coast (138) | 124/62 | 71/8 | 122/45 | 54/7 | 115/37 | 43/4 | 112/70 | 51/6 | 114/58 | 32/5 |
| III | Tasmania (25) | 21/13 | 0/0 | 23/11 | 0/0 | 20/10 | 0/0 | 21/7 | 0/0 | 21/9 | 0/0 |
| IV | Murray-Darling Basin (133) | 117/74 | 28/4 | 115/60 | 100/14 | 120/70 | 23/3 | 115/85 | 11/2 | 119/47 | 27/3 |
| V | South Australian Gulf (8) | 8/4 | 0/0 | 8/3 | 5/1 | 8/2 | 0/0 | 8/5 | 0/0 | 5/4 | 3/0 |
| VI | South West Coast (50) | 42/26 | 4/2 | 44/28 | 23/8 | 45/29 | 3/2 | 41/19 | 3/0 | 39/25 | 15/5 |
| VII | Indian Ocean (10) | 7/1 | 0/0 | 9/3 | 1/0 | 9/7 | 1/0 | 9/2 | 7/0 | 9/5 | 2/1 |
| VIII | Tanami-Timor Sea Coast (21) | 19/10 | 1/0 | 16/1 | 13/0 | 16/6 | 16/3 | 17/10 | 8/1 | 15/12 | 0/0 |
| IX | Carpentaria Coast (10) | 9/1 | 3/0 | 9/0 | 4/0 | 10/1 | 8/1 | 10/2 | 5/0 | 7/2 | 1/0 |
| X | Lake Eyre Basin (4) | 4/1 | 0/0 | 3/1 | 2/0 | 4/2 | 1/0 | 4/0 | 1/0 | 3/2 | 1/0 |
| XI | North Western Plateau (2) | 2/0 | 0/0 | 2/0 | 1/0 | 2/1 | 1/0 | 2/1 | 2/0 | 2/0 | 0/0 |
| XII | South Western Plateau (0) | --- | --- | --- | --- | --- | --- | --- | --- | --- | --- |
| | Total | 412/196 | 130/14 | 408/154 | 221/30 | 400/183 | 127/14 | 399/216 | 110/9 | 380/169 | 90/14 |


**Figure 6: Spatial distribution of stations where streamflow shows persistence, Short Term (STP) and Long Term (LTP), in (a) autumn (Mar-May), (b) winter (Jun-Aug), (c) spring (Sep-Nov), (d) summer (Dec-Feb), and (e) annual (water year) at p < 0.10.**

### 4.1.3 Linear trends - MK tests

Table 4 summarises the MK3, MK3bs and MK4 test results. Fig. 7. show the distribution of trends for each drainage division for the water year, autumn, winter, spring, and summer for all three MK tests. The MK3 test





results are similar to the MK3bs test results for water years and all four seasons, as they both consider the full autocorrelation structure (STP) of the streamflow series. Spatial distribution of trends under all three MK tests in the water year suggests the annual mean streamflow has increased in Tanami-Timor Sea Coast division in northern Australia and decreased in the southwest and southeast parts of the country (Fig. 7). Magnitude trends in streamflow volumes expressed by Theil-Sen's slope show a maximum of 2.4 %/year increase at one station in the

Tanami-Timor Sea Coast (VIII) and a maximum decrease of -3.9 %/year at one station in the Murray-Darling Basin (IV) over a period of at least 50 years. The MK3 test results are almost like that of MK3bs test for water years and all four seasons, as they both consider the full autocorrelation structure (STP) of the streamflow series. Therefore, MK3 test results are initially analysed.

Though most of all 467 streamflow stations showed trends, across water years, only 14 stations had data with increasing trends, and 212 stations showed decreasing trends that were statistically significant (Table 4). In the

water year, more than 50% of stations in the drainage division with increasing trends was the Tanami-Timor Sea Coast (VIII). Streamflows that showed decreasing trends were within the South East Coast (II), Tasmania (III), Murray-Darling Basin (IV), South Australian Gulf (V) and South West Coast (IV) drainage divisions. Most stations in other divisions showed no significant trend (Fig.7)

For autumn (Mar-May), streamflows at more than 50% of stations showed significant trends in the South East

Coast (II), Tasmania (III), Murray-Darling Basin (IV), South Australian Gulf (V) and South West Coast (IV) and North East Coast (I) divisions. Other divisions showed autumn decreasing trends that were very similar to those found across water years, except for the Tanami-Timor Sea Coast (VIII) in the north, where only one station showed an increasing trend. Data from 173 stations showed significant trends across water years, 4 out of which were increasing and 169 were decreasing (Table 4). In autumn, the maximum increase in streamflow was

1.3%/year, and the maximum decrease was -5.8%/year. Both trends were at stations in South West Coast (IV).

Across winter (Jun-Aug), streamflows at over half of the stations showed significant decreasing streamflow trends in the North East Coast (I), Tasmania (III), Murray-Darling Basin (IV), and South West Coast (IV) and significant increasing trends in the Tanami-Timor Sea Coast (VIII). However, in the South East Coast (II), Pilbara-Gascoyne (VII), Carpentaria Coast (IX) and the Lake Eyre Basin (X), fewer than 50 % of stations had

flows with significant trends. A total of 189 stations showed significant trends (for MK3) in streamflows across Australia, of which 9 are increasing, and 180 are decreasing (Table 4). The maximum increase in winter flows was 2.3 %/year (in the Tanami-Timor Sea Coast (VIII)) and the maximum decrease was -3.4 %/year (in the Murray-Darling Basin (IV). For stations in the Murray-Darling Basin (IV), significant trends vary between -0.5 % and -3.4 % (median of -1.3%/year).

Spring (Sep-Nov) saw more stations with decreasing flow trends compared with water years or the other seasons. Trends in flow were mostly detected at stations in all divisions except for Lake Eyre Basin (X) and North Western Plateau (XI), for which there is very limited flow data. South East Coast (II) and Murray-Darling Basin (IV) divisions had the highest number of stations with decreasing trends in streamflow, compared with trends in the water year and other seasons (Table 4). There were stations in the South East Coast (II), Tasmania

(III), Murray-Darling Basin (IV) and South Australian Gulf (V) that showed significant decreasing trends. For spring (Sep-Nov), 235 stations showed significant trends – 8 increasing and 227 decreasing (Table 4). The



maximum increase in spring flows was 3.1 %/year (in the Tanami-Timor Sea Coast (VIII)), and the maximum decrease was -2.8 %/year respectively (in the South West Coast (IV) divisions.

For summer (Dec-Feb), stations in the Tanami-Timor Sea Coast (VIII) showed significant increasing flow trends as with trends in other seasons. Stations in the South East Coast (II), Tasmania (III), Murray-Darling Basin (IV), South Australian Gulf (V) and South West Coast (VI) divisions showed significant decreasing trends in flow. In summer, 175 stations showed significant flow trends – 26 increasing and 149 decreasing (Table 4). The maximum increase in summer flows was -2.1%/year (in the Tanami-Timor Sea Coast (VIII)) and the maximum decrease was -5.0%/year (in the South West Coast (IV) division.

While results from the MK3 and MK3bs tests are nearly identical for most cases, MK3 and MK4 do show differences for most annual and seasonal flow statistics. A typical example of MK3bs statistic obtained from 2000 samples of two locations with a) decreasing and b) increasing trends is shown in Fig. 8. A small number of stations have significant trends in streamflow when LTP behaviour (MK4) was considered (Fig. 7, Table 4). This is particularly evident in South East Coast (II) for water years, and in the Murray-Darling Basin (IV) for
autumn (Table 4, Fig. 7). The MK4 test resulted in more stations with trends in winter and spring, compared to autumn and summer. Across Australia for water years, 14 stations showed significantly increasing trends in flow, and 196 stations showed significantly decreasing trends, slightly less than MK3 and MK3bs test results (Table 4). Similar results are also evident for four seasons across Australia (Table 4).




Table 4 Results of three Mann Kendall (MK) tests for water year and all four seasons

| Drainage division | Drainage division name | Water year | | | Autumn | | | Winter | | | Spring | | | Summer | | |
| | | MK3 +/- | MK3bs +/- | MK4 +/- | MK3 +/- | MK3bs +/- | MK4 +/- | MK3 +/- | MK3bs +/- | MK4 +/- | MK3 +/- | MK3bs +/- | MK4 +/- | MK3 +/- | MK3bs +/- | MK4 +/- |
|---|---|---|---|---|---|---|---|---|---|---|---|---|---|---|---|---|
| I | North East Coast | 0/4 | 0/9 | 0/2 | 1/2 | 0/2 | 0/2 | **3/10** | **2/11** | **2/9** | **3/11** | **2/12** | **2/11** | 0/6 | **0/6** | 0/6 |
| II | South East Coast | **0/64** | **0/64** | **0/54** | **0/58** | **1/60** | **0/40** | 0/42 | **0/38** | **0/41** | **0/77** | **0/77** | **0/69** | **0/58** | **0/58** | **0/50** |
| III | Tasmania | **0/15** | **0/13** | **0/15** | **0/16** | **0/15** | **0/16** | **0/12** | **0/13** | **0/13** | 0/8 | **0/7** | **0/7** | **0/9** | **0/9** | **0/9** |
| IV | Murray-Darling Basin | **0/89** | **0/82** | **0/84** | **0/61** | **0/67** | **0/32** | **0/75** | **0/76** | **0/70** | **0/98** | **0/92** | **0/93** | 1/48 | **0/46** | **0/42** |
| V | South Australian Gulf | **0/5** | **0/3** | **0/5** | **0/3** | 0/4 | 0/2 | **0/3** | 0/3 | 0/3 | **0/5** | 0/3 | 0/5 | **1/5** | 0/4 | **0/4** |
| VI | South West Coast | **0/33** | **0/33** | **0/34** | 1/27 | **3/28** | **1/26** | **0/33** | **0/32** | **0/33** | 1/22 | **1/23** | **1/21** | **5/22** | **5/20** | **5/21** |
| VII | Pilbara-Gascoyne | 1/1 | 1/1 | 1/1 | 1/1 | 1/1 | 1/1 | 0/4 | **0/5** | **0/4** | 1/2 | 1/2 | 0/0 | 2/2 | 2/0 | 2/0 |
| VIII | Tanami-Timor Sea Coast | **12/0** | **11/0** | **12/0** | 1/0 | 1/0 | 0/0 | **5/0** | 6/0 | 4/0 | 2/4 | 4/1 | 2/1 | **14/0** | 13/0 | **13/0** |
| IX | Carpentaria Coast | 1/0` | 1/0` | 1/0` | 0/0 | 1/0 | 0/0 | 1/1 | 1/1 | 1/0 | 1/1 | 1/1 | 0/1 | 2/0 | 2/0 | 2/0 |
| X | Lake Eyre Basin | 0/1 | 0/1 | 0/1 | 0/1 | 0/1 | 0/0 | 0/1 | 0/1 | 0/1 | 0/0 | 0/0 | 0/0 | 1/1 | 1/1 | 1/0 |
| XI | North Western Plateau | 0/0 | 0/0 | 0/0 | 0/0 | 0/0 | 0/0 | 0/0 | 0/0 | 0/0 | 0/0 | 0/0 | 0/0 | 0/0 | 0/0 | 0/0 |
| XII | South Western Plateau | --- | --- | --- | --- | --- | --- | --- | --- | --- | --- | --- | --- | --- | --- | --- |
| | Total | 14/212 | 13/206 | 14/196 | 4/169 | 7/178 | 2/119 | 9/180 | 9/180 | 7/174 | 8/227 | 9/218 | 5/208 | 26/149 | 23/144 | 23/131 |

MK3, MK3bs, MK4 correspond to MK tests

+ Number of stations showing increasing trends

- Number of stations showing decreasing trends

Entries in bold indicate results that are field significant at p < 0.10.


**Figure 7: Maps showing seasonal trends using (a) MK3, (b) MK3bs, and (c) MK4 tests. Results are reported for autumn, winter, spring, summer, and annual (water year) (p < 0.1). Upward-pointing triangles (green) indicate significant increasing trends, and downward-pointing triangles (red) indicate significant decreasing trends. Grey dots indicate stations with no trends. Drainage divisions with positive and negative trends with field significance at p < 0.10 are coloured blue and yellow, respectively.**

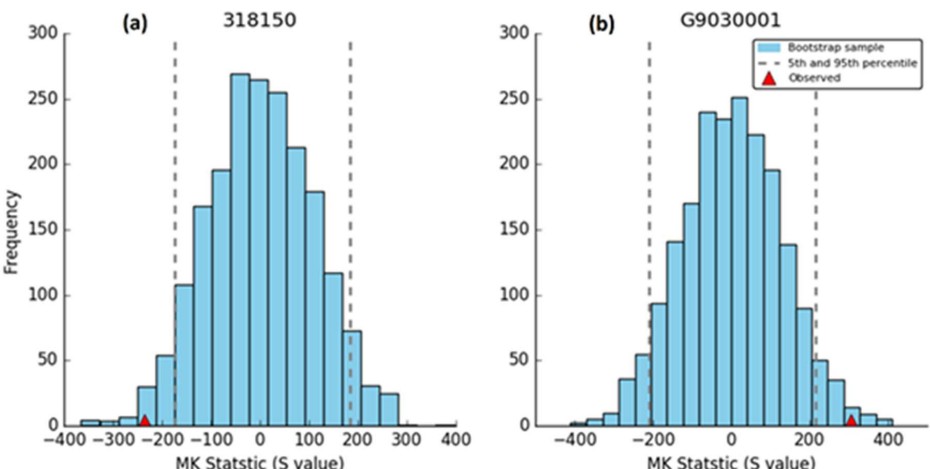

**Figure 8: Examples of histograms for two stations representing the frequency distribution of MK Statistic**

**(S) obtained from 2000 samples of moving-block bootstrap iterations; red triangles are observed values,**
**while dotted lines show the 5th and 95th percentiles for (a) decreasing trend (318150) and (b) increasing**
**trend (G9030001) in Tasmania (III) and Carpentaria Coast (IX) divisions, respectively**

### 4.2    Step change

The non-parametric Pettitt test (Pettitt, 1979) was used to test for step changes for water years as well as for all

seasons. It is least sensitive to outliers, and skewed distribution makes it most suitable for the analysis of
streamflow data. A typical example of step change in northern Australia in shown in Fig. 5b. Step change maps
(Fig. 9a) clearly reveal a spatial pattern in the location of stations that exhibited a significant step change in
flow. The direction and significance of step changes are consistent with results from MK tests (Fig. 7) for most
stations. Years when step changes occurred show spatial groupings within several drainage divisions.

Significant step changes or shifts across water years, and all four seasons for each drainage division are
summarised in Table 3.  For water years, the South East Coast (II), Murray-Darling Basin (IV), South Australian
Gulf (V), South West Coast (VI) and Tanami-Timor Sea Coast (VIII) drainage divisions all showed significant
step changes for more than 60% of station flows (Fig. 9a, Table 3). Increasing shifts were seen in northern
Australia, including the Indian Ocean (VII), Tanami-Timor Sea Coast (VIII) and Carpentaria Coast (IX)

divisions, whereas decreasing shifts were seen in all drainage divisions except for the Tanami-Timor Sea Coast
(VIII), Carpentaria Coast (IX) and North Western Plateau (XI). The South East Coast (II), Murray-Darling
Basin (IV), South West Coast (VI) and Tanami-Timor Sea Coast (VIII) had significant step changes across
water years (Fig. 9).

For summer (Dec-Feb), the lowest proportion of stations had significant step changes in flow compared with the

other seasons or with water years (Table 3). The South East Coast (II), Murray-Darling Basin (IV) and South
West Coast (VI) divisions had significant step changes at $p < 0.10$ for water years and for all seasons, while



Tanami-Timor Sea Coast (VIII) step changes for water years and all seasons except for autumn. Tasmania (III) had significant step changes only in autumn. North East Coast (I) had significant step changes in spring and winter. South Australian Gulf (V) had significant step changes in autumn and winter (Fig. 9).

Figure 9 shows the number of stations step changes in flow, each year between 1950 and 2018 for water years, and for each season. The first step change was detected in 1964, and most changes occurred between 1970 and 1999. Out of 467, a total of 253 stations show step changes in flows for water years (p < 0.10) of which 16 were increasing and 237 were decreasing changes in flows. Water years between 1992 and 2004 had increasing step changes in 16 stations, with the water-year 1996 having 9 stations with increasing step changes in flow

(Fig. 9b). The period from 1975 to 1979 showed decreasing step changes for 24 stations; out of which 18 were in the 1978 water year. The period from 1981 to 1989 show decreasing step changes for 8 stations, and the period from 1990 to 2000 showed decreasing step changes in 190 stations, half of which occurred in the 1996 water year.

Autumn (Mar-Jun) had a total of 210 stations with step changes (p < 0.10) in flow, of which 8 were increasing

and 202 decreasing. The first increasing step change in autumn was detected in 1970 (Fig. 9b), in the Carpentaria Coast (IX) division. Similar to water years, most step changes occurred between 1990 and 2000, with 147 stations showing decreasing step changes in flow (Table 5).  About quarter of the step changes took place in the 1996 water year.

In winter (Jun-Aug) 246 stations showed a step change (p < 0.10) in flow, of which 16 were increasing and 230

decreasing. The first step change in winter occurred in 1970 (Fig. 9b). One station showed increasing shifts starting early in the 1970s, and 10 additional stations during 1996 to 1999. During the period from 1975 to 1982, 10 stations showed decreasing step changes in flows. Between 1990 and 2000, 211 stations (the largest in total), showed a decreasing step change in flows, with 101 of these changes occurring in 1996.

Spring (Sep-Nov) had a total of 219 stations with step changes (p < 0.10) in flow, of which 21 stations had an

increasing step change and 198 stations had a decreasing shift. The period from 1994 to 1997 saw 11 stations with increasing step changes (Fig. 9b) and in the period from 1989 to 2001, 178 stations had decreasing step changes in their spring flows.

Summer had a smaller number of stations (160 out of 467) with step changes (p < 0.10) in flows, of which 24 stations had increasing shifts and 136 stations had decreasing step changes (Table 3, Fig.9b). The distribution of

stations with step changes varied across the period of record; with 28 stations with decreasing step changes during 1976-1989, 4 stations increasing during 1980-1988, 101 stations decreasing during 1992-2002 and 18 station increasing during 1990-2000 (Fig. 9b).

Fig. 10 shows the timing of step changes for several drainage divisions the water-year range mostly from 1976 to 2009. The period from 1990 to 2000 showed decreasing step changes for 199 stations, mostly in the South

East Coast (II), Murray-Darling Basin (IV) and South West Coast (VI) divisions. Most increasing step changes are in the Tanami-Timor Sea Coast (VIII) division during 1992-1998. The South East Coast (II), Murray-Darling Basin (IV), South West Coast (VI) and Tanami-Timor Sea Coast (VIII) divisions showed a greater number of stations with step changes distributed across the study period, indicating a higher association to natural changes and climate variability than for other regions (Table 3).





After the step changes, the distribution of mean monthly streamflow within a water-year has changed across all drainage divisions. Distribution of mean monthly streamflow at four selected gauging stations from four drainage divisions are shown in Fig.11. In northern part of Australia, the increase in streamflow is well-distributed over most of the months. However, in south-west Queensland and southern part of Australia, there seems to be a phase shift as well.

Table 5. Annual and seasonal shifts in different drainage divisions using Pettitt test at p <0.10

| Drainage division | Drainage division (no. of stations) | Stations showing shifts | | | | |
|---|---|---|---|---|---|---|
| | | Water-year +/- | Autumn +/- | Spring +/- | Summer +/- | Winter +/- |
| I | North East Coast (66) | 0/10 | 0/3 | **4/14** | 0/5 | **2/21** |
| II | South East Coast (138) | **0/83** | **0/69** | 3/72 | 0/56 | **0/58** |
| III | Tasmania (25) | 0/9 | **0/15** | 0/0 | 0/7 | 0/9 |
| IV | Murray-Darling Basin (133) | **0/96** | **0/81** | 0/81 | 1/41 | **0/98** |
| V | South Australian Gulf (8) | 0/5 | **0/5** | 0/3 | 1/5 | **0/5** |
| VI | South West Coast (50) | **0/32** | 4/27 | 0/20 | 5/21 | **0/34** |
| VII | Pilbara-Gascoyne (10) | 2/1 | 0/1 | 0/1 | 2/0 | 0/3 |
| VIII | Tanami-Timor Sea Coast (21) | **13/0** | 2/0 | **11/21** | **13/0** | **12/0** |
| IX | Carpentaria Coast (10) | 1/0 | 2/0 | 0/2 | 2/0 | 2/0 |
| X | Lake Eyre Basin (4) | 0/1 | 0/1 | 0/1 | 0/1 | 0/2 |
| XI | North Western Plateau (2) | 0/0 | 0/0 | 0/0 | 0/0 | 0/0 |
| XII | South Western Plateau (0) | -- | -- | -- | -- | -- |
| Total (467) | | 16/237 | 8/202 | 21/198 | 24/136 | 16/230 |

+ Number of stations showing increasing shifts

- Number of stations showing decreasing shifts

Entries in bold indicate results that are field significant at p < 0.10.



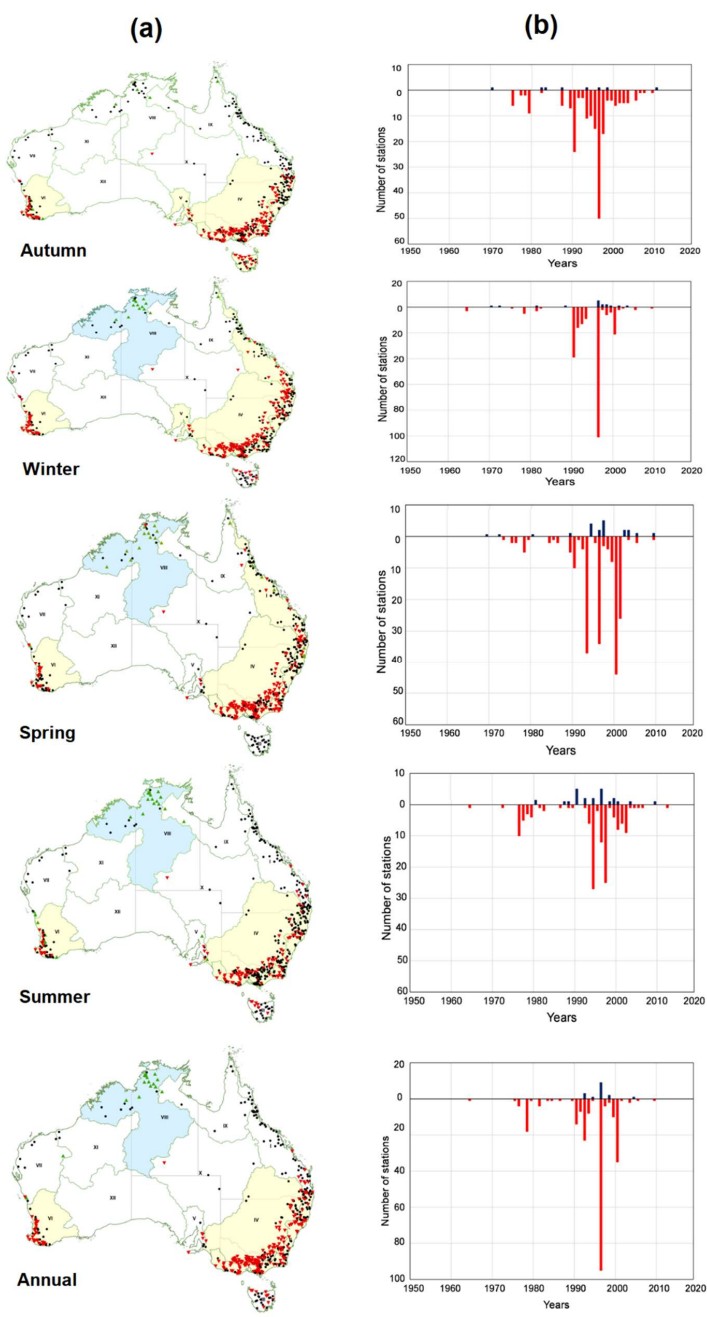

**Figure 9: (a) Map showing stations with step changes in flow across each season and water year (at significance level p < 0.1), upward pointing green triangles and downward pointing red triangles represent upward and downward step changes, respectively, while black dots refer to sites without**




significant shifts. Drainage divisions with positive and negative shifts with field significance at p < 0.1 are coloured blue and yellow, respectively. and (b) Number of stations with significant step changes in flows (p < 0.1) across each season and water years.


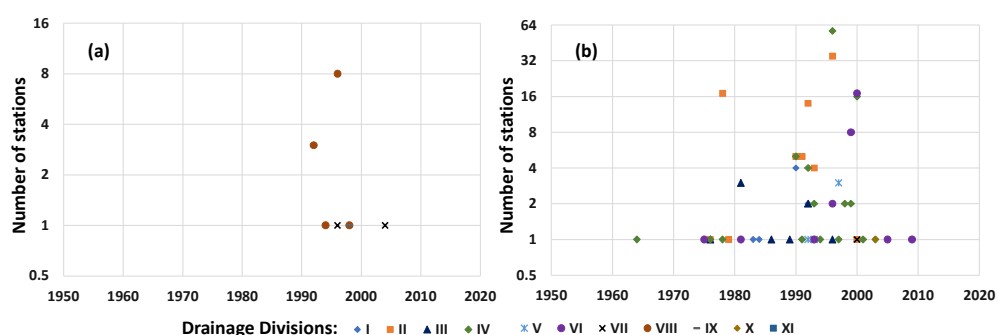

**Figure 10: Timing of, and number of stations with step changes for different drainage divisions with (a) increasing and (b) decreasing changes in the water year.**


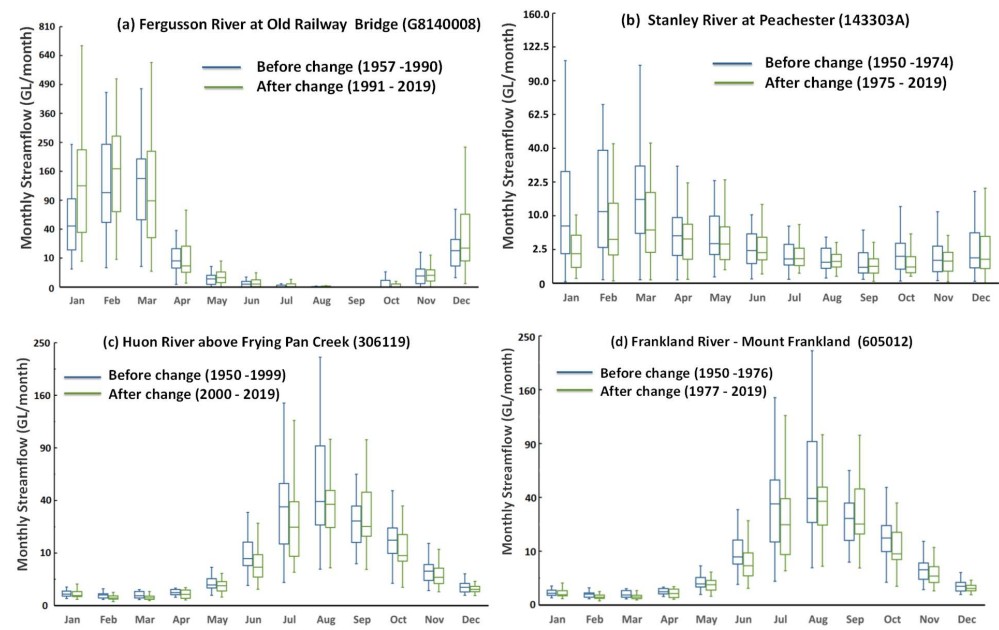

**Figure11: Distribution of mean monthly streamflow before and after step changes at four selected gauging stations representative of (a) Tanami-Timor Sea Coast (VIII), (b) North East Coast (I), (c) Tasmania (III) and (d) South West Coast (VI). See Fig. 1 for locations.**

**4.3    Trend summary of regional significance – drainage divisions**



Table 6 summarises field or regional significance at drainage division-scale of linear trends and step changes within each drainage division for water year and each of the four seasons. The Murray Darling Basin (IV), South East Coast (I) and South West Coast (VI) divisions experienced decreasing trends for water year, and for all seasons across all statistical tests. Similarly decreasing trends were also evident in the North East Coast (I), Tasmania (III) and South Australian Gulf (V) divisions with the exception of only a few statistical tests (Fig. 7, Fig. 9). In the North East Coast (I) division, only the MK3 test detected no significant annual trends. In the Tasmania (III) division, no step changes (PT) or linear trends (Mk4) were detected for spring and summer, respectively. In northern Australia, Tanami- Timor Sea Coast is the only division where most of the statistical tests show increasing trends for water years and for all seasons (Table 6). At annual scales, MK3bs, Mk4 and PT tests showed significant increasing trend. With the exception of autumn, MK3 and PT tests detected increasing linear trends for the other three seasons. There were no noticeable patterns in divisions in central Australia, including North Western Plateau (XI) and Lake Eyre Basin (X) divisions. Trends in the Carpentaria Coast (IX) were varied, and only summer saw a significant decreasing trend in flows (MK3). The lack of streamflow observations and relatively low number of stations, or length of records, or number of non-zero flow days in these central Australian divisions may play a role in detecting the trend.



Table 6 Summary of field significance across different drainage divisions - water year and four seasons

| Drainage Division | | Water Year | | | | Autumn | | | | Winter | | | | Spring | | | | Summer | | | |
|---|---|---|---|---|---|---|---|---|---|---|---|---|---|---|---|---|---|---|---|---|---|
| No. | Name | MK3 | MK3bs | MK4 | PT | MK3 | MK3bs | MK4 | PT | MK3 | MK3bs | MK4 | PT | MK3 | MK3bs | MK4 | PT | MK3 | MK3bs | MK4 | PT |
| I | North East Coast | -- | ✓ | ✓ | | | | | | ✓ | ✓ | ✓ | | ✓ | ✓ | ✓ | ✓ | | | | |
| II | South East Coast | ✓ | ✓ | ✓ | ✓ | ✓ | ✓ | ✓ | ✓ | | ✓ | ✓ | ✓ | ✓ | ✓ | ✓ | ✓ | ✓ | ✓ | ✓ | ✓ |
| III | Tasmania | ✓ | ✓ | ✓ | ✓ | ✓ | ✓ | ✓ | ✓ | ✓ | ✓ | ✓ | ✓ | ✓ | ✓ | ✓ | | ✓ | ✓ | ✓ | |
| IV | Murray-Darling Basin | ✓ | ✓ | ✓ | ✓ | ✓ | ✓ | ✓ | ✓ | ✓ | ✓ | ✓ | ✓ | ✓ | ✓ | ✓ | ✓ | | ✓ | ✓ | ✓ |
| V | South Australian Gulf | ✓ | ✓ | ✓ | | ✓ | ✓ | ✓ | ✓ | ✓ | ✓ | ✓ | ✓ | | | ✓ | | ✓ | ✓ | ✓ | |
| VI | South West Coast | ✓ | ✓ | ✓ | ✓ | ✓ | ✓ | ✓ | ✓ | ✓ | ✓ | ✓ | ✓ | ✓ | ✓ | ✓ | ✓ | ✓ | ✓ | ✓ | ✓ |
| VII | Pilbara-Gascoyne | | ✓ | | | | ✓ | | | | ✓ | | | | | | | | | | |
| VIII | **Tanami-Timor Sea Coast** | ✓ | ✓ | ✓ | ✓ | ✓ | | ✓ | ✓ | | ✓ | ✓ | ✓ | | | ✓ | ✓ | ✓ | ✓ | ✓ | ✓ |
| IX | Carpentaria Coast | | | | | | | | | | | | | | | | | | | | |
| X | Lake Eyre Basin | | | | | | | | | | | | | | | | | | | | |
| XI | North Western Plateau | | | | | | | | | | | | | | | | | | | | |
| XII | South Western Plateau | -- | -- | -- | -- | -- | -- | -- | -- | -- | -- | -- | -- | -- | -- | -- | -- | -- | -- | -- | -- |

MK3, MK3bs, MK4 correspond to MK tests

PT corresponds to Pettitt Test

Entries in bold indicate positive upward trend at p < 0.10.




## 5 Discussion

### 5.1 Data quality and gap-filling

To maintain the quality of streamflow data, only gauging stations with less than 25% of flow volume above the recorded maximum gauged discharge and less than 10% of infilled flow volume were included in this study and Hydrologic Reference Stations. A quantitative evaluation of the accuracy of gap-filling procedure showed that gap-filling by interpolation or simple rainfall-runoff modelling is most accurate and effective if the missing record is less than 10% (Zhang and Post, 2018), which was followed in this study. However, when the percentage of gap-filled data represent relatively high end of the flow volume, the uncertainty becomes higher.

McMahon and Peel (2019) analysed uncertainties associated with streamflow by considering the gauged and extended range of the stage–discharge relationship from 622 rating curves for 171 of the HRS gauging stations. They found that estimated flow volumes beyond the gauged range of the rating curve occurred in many of these gauging stations and caused measurement uncertainty. As our analysis is limited to seasonal and annual totals, implications of the estimated volumes beyond the gauged range may be minimal.

### 660 5.1 Comparison of different tests

The MK3 and MK4 tests show there are strong autocorrelations or short-term persistence (STP) and significant long-term persistence (LTP) respectively in Australian streamflow in most drainage divisions (Table 3, Figs. 6). About 85% of stations had data with significant autocorrelation structure (Table 3), and about 40% of these had more than lag-1 autocorrelation (Fig. 3). As such, the MK2 test (Kumar et al.,2009, Su et al.,2018) was not

considered in the analysis because it considers only the lag-1 autocorrelation structure. Around 30% of stations had significant LTP only (Table 3). Therefore, two variations of MK3 which considers the total autocorrelation structure (STP) and MK4 which considers LTP behaviour were used in the analysis. Our calculated trends are strongly affected by STP and LTP factors. For water years, there was a clear reduction in number of stations with significant streamflow trends when the full autocorrelation structure (MK3) and LTP behaviour (MK4)

were considered in the analysis: from 263 stations with MK1 to 226 with MK3 and 210 with MK4. A similar reduction was observed for most of the seasons (Table 4). Around half of stations with STP and only about 12% of stations with LTP had significant trends for water year as well for all four seasons. The application of these different MK tests helped to differentiate trends that exist under the assumption of serial independence and short and long-term persistence.

The Pettitt test showed 54% of stations experienced an abrupt shift in streamflows across water year. Most of these step changes (51%) were decreases and were observed in more stations than for linear decreasing trends (38%). Most of the increasing step changes took place in spring (Sep-Nov) and summer (Dec-Feb), whereas most of the decreasing step changes occurred in winter (Jun-Aug). 1970 to 1999 had the largest number of step changes for water year flows. The greatest number of stations (21%) with a decreasing step change was in 1996,

whereas most of increasing shifts (2%) occurred in 1996. Seasonally, a greater number of stations with significant decreasing step changes in flow occurred in winter (Fig. 9b-10).





### 5.3 Attribution of trends

For annual time scales, linear and step changes were similar in direction across the country, however slight
variation in direction was observed between seasons and different river basins. These are consistent with
previous findings for Australian streamflows (Zhang et al, 2016) and other studies despite having larger number
of gauge locations in all drainage divisions except South West Plateau, different analyses methods, length and
coverage period of data (Durrant and Byleveld, 2009; Petrone et al., 2010; Tran and Ng, 2009). These findings
are consistent with observed changes in rainfall in different parts of Australia (CSIRO and BOM, 2020) where i)
Northern Australia has become wetter, particularly in the northwest, with rainfall generally above average in the
dry seasons; ii) Australia's climate has warmed on average by about 1.5°C since national records began in 1910,
leading to an increase in frequency of extreme heat events and in the length of the fire season, across large parts
of the country since the 1950s, especially in southern Australia; iii) For April to October rainfall, there has been
a decline of around 16% in the southwest of Australia since 1970 and a decline of around 12% in the southeast
of Australia since the late 1990s; iv) Rainfall and streamflow have increased across parts of northern Australia
since the 1970s (State of the Climate 2020). In general, observed patterns of rainfall change across the country
are expected to continue (CSIRO and BOM, 2020). Further research is required to reveal the association of
historical rainfall changes with observed streamflow across different drainage divisions of Australia. Data from
these national network of hydrologic reference stations will be useful for examining and detecting these
associations.

Linear changes in streamflow generally may occur in response to long-term changes in the flow generation
processes including increases or decreases in rainfall, evapotranspiration, temperature, humidity and changes in
vegetation dynamics. Note that the catchments considered here are unimpaired, so the impact of anthropogenic
activities are not significant. For some of the locations, the decreasing step changes coincide with the
Millennium Drought between 1997 to 2009, some catchments underwent changes a few years before that period
(Fig. 14). Streamflow deficits in the Murry-Darling Basin were very large during the drought, with an estimated
return period of 1 in 1500 years (Gergis et al., 2012), which may be one of the reasons for the largest proportion
of stations (96 out of 133) depicting step changes. Sudden and abrupt changes including step changes in
streamflow generation process happen due to changes in thresholds which is most prominent when the filling
season begins – autumn (Mar-Jun) in south west of Australia (Silberstein et al., 2012) including Tasmania
(Fig 9, Table 6). Also evident in phase shift of streamflow distribution in south-west of Western Australia
(Fig. 11d), due to reduction in rainfall and lateral shift within the year (Silberstein et al., 2012) . Reduction in
winter rainfall, increase in temperature and extreme rainfall are projected to continue in the future (CSIRO and
BOM, 2020). Big questions remain around how these changes in driving climate forces will interact with each
other, and with vegetation, to guide the types and rates of change in streamflow trend.

### 5.4 Management and further research

The frequency and magnitude of extreme rainfall events have increased across Australia (Sharma et al., 2018)
and further investigation is required to understand how this translates to maximum flow and flood extremes in
different parts of the country. Changes in the frequency and duration of low flows are also if interest to water





managers, particularly in maintaining environmental water flow requirements. Questions remain in relation to
phase shifts of streamflow generation within the year (Fig. 11), in particular southern part of Australia. Are they
related to rainfall distribution alone, or are there other contributing factors, such as changes in temperature or
secondary changes in vegetation dynamics? The influence of step changes on gradual trends was not considered
in this study. Further investigation to identify the conditions and processes that result in these step changes
would be of great value. Research into these questions is important to guide better management of water
resources, infrastructure development and environmental water allocations. The statistical analysis undertaken in
this study depends on the length of the time series available, which varies one station to the next (Fig. 1). Better
understanding of the nature of trends and step changes in seasonal streamflow can support better regional water
management to regulate the flows and maintain adequate levels in reservoirs during dry and wet periods.

## 6    Summary and Conclusions

Trends in annual and seasonal streamflow over Australia at 467 high quality gauging stations listed in Bureau of
Meteorology's Hydrologic Reference Stations were analysed and presented. Length of the daily streamflow data
record ranged from a minimum of 30 years to a maximum of 69 years. Linear trend analyses were performed
using three different forms of Mann Kendall tests (Variance Correction Approach – MK3, Block Bootstrap
Approach – MK3bs and Long-Term Persistence – Mk4). Identification and detection of step changes for
seasonal and annual streamflow were accomplished by using nonparametric Pettitt test. Regional significance at
different drainage division scale of these changes was analysed and synthesised by using Walker test.

Linear decreasing trends in annual and seasonal streamflow at most of the gauging stations were detected in the
Murray-Darling River basin and other drainage divisions in New South Wales, Victoria and Tasmania. Similar
results were observed in the south-west of Western Australia, South Australia and in north-east Queensland.
Decreasing trends in annual totals and in totals for all four seasons were also regionally significant at the
drainage division scale. Only the Tanami-Timor Sea Coast drainage division in northern Australia showed
increasing trends and step changes in annual and seasonal streamflow, and these were regionally significant.
There were no significant spatial and temporal patterns observed in central and mid-west Australia. One
possibility for this is the sparce density of streamflow stations and length of data available for analysis.

In general, step changes were similar to the direction of linear trends across Australia. Only a handful of stations
in northern Australia showed significant step change increases in streamflow. At regional scales, that increasing
trend was statistically significant at only the Tanami-Timor Sea Coast drainage division. Across southern
Australia, most step changes occurred during 1970-90s, majority of them being in 1990s, before the onset of
millennium drought in 1997. Most of the step changes occurred just at the start of winter when the rainy season
begins.

Further investigation and research would assist to understand the processes that govern the detected changes in
flow generation, catchment memory and its interaction with rainfall change, increase in temperature and
vegetation growth and evapotranspiration.



**7      Acknowledgements**

Streamflow data was initially provided by national and state water agencies across Australia. The Hydrologic Reference Stations website was developed in consultation with University of Melbourne, CSIRO Land and Water, Department of Climate Change and Energy Efficiency (DCCE) and approximately 70 other stakeholders. We thank Emeritus Professor Tom A. McMahon for his ongoing contributions to the HRS technical review. We

would also like to express our sincere thanks to the editor, Dr Julien Lerat, Dr Margot Turner for their time, careful review and valuable comments and suggestions on the submitted version of this paper.

**Author Contributions**

Amirthanathan, G.E. undertook to data curation, formal analyses, investigation, methodology, validation,
visualisation and writing. Bari, M.A contributed to conceptualisation, investigation, methodology, project administration, resources, supervision, validation and writing. Woldemeskel, F. contributed to data curation, formal analyses, investigation, methodology, validation and visualisation. Feikema, P.M provided project administration, resource allocation, supervision, validation, manuscript review and editing.



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
