# Peer review of "Regional significance of historical trends and step changes in Australian streamflow"

_Hydrology and Earth System Sciences, 2022_

## Author Response (AR1)

Hydrol. Earth Syst. Sci. Discuss., referee comment RC1
https://doi.org/10.5194/hess-2022-199-RC1, 2022

[Figure]

**Comment on hess-2022-199**

Anonymous Referee #1
* * *
Referee comment on "Regional significance of historical trends and step changes in Australian streamflow" by Gnanathikkam Amirthanathan et al., Hydrol. Earth Syst. Sci. Discuss., https://doi.org/10.5194/hess-2022-199-RC1, 2022
* * *
**(Note: All line numbers, figures, tables and sections refer to the clean copy of the manuscript)**

**Summary:**

**Reviewer comment**: This manuscript uses statistical tests to show the historical trend of the streamflow in Australian, which is intriguing. The results would be useful for the future study to learn how climate change, evapotranspiration, rainfall pattern or other factors affect the streamflow pattern.

*Author response: We thank the reviewer for their appreciation and acknowledgement of the results of our paper. We addressed all the comments by the reviewer as described below.*

**Major Comments:**

**Reviewer comment**: For several statistical tests, the 0.1 of P value were used. Nowadays, many research use p value of 0.05, could you give us more detail or explanation why 0.1 has been used.

*Author response: We thank the reviewer for asking for an explanation of using p values of 0.10 instead of 0.05. While a p value of 0.05 or less would be preferable for a strong statistical significance test, a p value of 0.1 has also been commonly used for a similar studies (e.g. Sagarika, et al., 2014). Further, in our previous paper, published on the Australian Hydrologic Reference Stations, for all the flow parameters, a p value of 0.1 was used except for the annual*

*total flow, where p values of 0.01, 0.05 and 0.10 was considered (Zhang, et al,.2016). In this paper, we decided to use p value of 0.1 for all the analyses, considering the brevity, consistency and content and size of the manuscript.*

**Reviewer comment**: Line 675, "abrupt shift in streamflows across water year", it would be better to show how abrupt percentage is based on the region (spatial) and year (year), like in Table.

*Author Response: This is an interesting suggestion and we have investigated this further. We have calculated the % change in annual total volume (average of the number of catchments within a division), as shown in the table below. The result presented in the table is for the editor/reviewer benefit only and we do not intend to include it in the paper for two reasons. First, when the mean change in streamflow is presented in this way, it may suggest that the change represents the whole region (division), however in reality the change is only on those HRS stations where a change was detected. Second, the respective year of a detected step change (for a specific station) varies, so it can be misleading to refer to a given mean step change, as this will have occurred over several years, and is not as abrupt as suggested.*

*Table X. Annual total volume shifts in different drainage divisions*

| Drainage division | Drainage division | Water-year | | Median year of shift** |
| --- | --- | --- | --- | --- |
| | | % Shift | | |
| | | Mean drop(-)/rise(+) | Std. deviation | |
| I | Northeast Coast (10/66)* | -45% | 8% | 1990 |
| II | Southeast Coast (83/138)* | -49% | 13% | 1992 |
| III | Tasmanian (9/25)* | -33% | 10% | 1986 |
| IV | Murray-Darling (96/133)* | -54% | 16% | 1996 |
| V | South Australia Gulf (5/3)* | -52% | 9% | 1997 |
| VI | Southwest Coast (32/50)* | -54% | 11% | 2000 |
| VII | Indian Ocean (1/10)* | -85% | -- | 2000 |
| VIII | Timor Sea (12/21)* | 77% | 21% | 1996 |
| IX | Gulf of Carpentaria (1/10)* | 120% | -- | 1998 |
| X | Lake Eyre (1/4)* | -89% | -- | 2003 |
| XI | North Western Plateau (0/2)* | -- | -- | -- |
| XII | South Western Plateau | -- | -- | -- |

*Note: ( )* refers to (number of sites with sig. shift/total number of sites in the division). These %change only represent the average of the sites and do not represent the basin.*
*** The median year of shift is given for sites with significant shifts and the year of shift varied within each drainage divisions.*

**Reviewer comment**: How does the long drought or extreme event affect the statistical test, like linear trend? E.g. after the long drought years, will it become increasing trend?

*Author Response: We thank the reviewer for this question. However, we used Theil–Sen estimator to compute the trend of the slope, which is generally insensitive to outliers. Therefore, this estimation of trend slope is not affected by extreme events (outliers). However, the level of statistical significance derived from performing MK test may be impacted due to extremes. Investigating the effect of long drought and extreme events on the statistical tests is an interesting and intriguing question and needs further investigation, which we consider out of scope of this paper.*

**Reviewer comment**: I recommend have a table or paragraph to show the raw data format like time interval, and explain how to deal with the raw data (take out extreme abnormal data, delete zero value). This is really important input for the following statistical tests.

*Author Response: We thank the reviewer for asking the details of data quality control process used in this study. Most of the gauging stations, included in the Hydrologic Reference Stations, are of high quality, with generally less than 5% missing data across the entire record. A quality-assurance, quality control (QA/QC) process was applied to observed time-series of daily streamflow from each gauging station. This process identified and removed erroneous data values such as negative and outliers, and periods of long linear interpolation. The process of detection and removal was automated and then checked manually. The website (http://www.bom.gov.au/water/hrs/references.shtml :Hydrologic Reference Stations update 2020) has details about this process. However, we have now added a sentence referring to this information available in the website – Line 174-175 .*

> The process of detection and removal was automated and then checked manually, as detailed in Hydrologic Reference Station website (http://www.bom.gov.au/water/hrs/references.shtml).

**Reviewer comment**: Climate change could make extreme events more often, but it could be possible that the average yearly precipitation will not change. In this case, it might have increasing trend for the wet season, decreasing trend for the dry season and no trend for the annual basis. Do you find any station having this similar situation?

*Author Response: We thank the reviewer of this interesting hypothesis and question. It is possible that seasonal and annual streamflow data from some stations may show these trends. However, we have not analysed data to test this hypothesis and this warrants further research. We consider this to be out of scope for this paper. However, we have added a short discussion of this issue in Section 5.3 and identify it as future research (Section 5.4). Line 715-716.*

Climate change could make extreme events more often – increasing streamflow trend for the wet season, decreasing trend for the dry season and no trend for the annual basis.

**Minor Comments:**

**Reviewer comment**: Line 39, please add reference to support the statement

*Author Response: We thank the reviewer for noting this point. We have included a new reference (Bureau of Meteorology, 2022).*

BoM (Bureau of Meteorology), 2022. Average annual, seasonal and monthly rainfall, Commonwealth of Australia. Accessed at:
http://www.bom.gov.au/jsp/ncc/climate_averages/rainfall/index.jsp

**Reviewer comment**: Line156, 23,2846 km separator is wrong. It should be corrected to

*Author Response: Apology for the typo error, now fixed > 232,846*

**Reviewer comment**: Figure 1, north arrow is missing

*Author Response: Apology of the omission, it is now fixed, please see below. The updated figure will be included in the manuscript.*

[Figure]

**Reviewer comment**: Line 409, how do you convert GL/year to mm/year:

*Author Response: We used a formula to convert GL to mm > 1 GL/year = (1000/A) mm/year; where A = catchment area in km². We will mention this in the manuscript.*

**Reviewer comment**: In section 4.2, the authors mention the Figure 3 (*Table 3*) several times. Did you mean by Figure 5 (*Table 5*)?

*Author Response: Thanks for this comment. We have closely looked at all refences (for both Tables and Figures) and found that Table 5 was wrongly refenced as Table 3 several times. We have now corrected this. However, we could not find Figure 3 or 5 mentioned wrongly.*

**Reviewer comment**: Line 706, where is the Figure 14?

*Author Response: Apology for the typo error. It should refer to Figure 10, this has now been fixed.*

**Comment on hess-2022-199**

Nir Krakauer Referee #2

Referee comment on "Regional significance of historical trends and step changes in Australian streamflow" by Gnanathikkam Amirthanathan et al., Hydrol. Earth Syst. Sci. Discuss., https://doi.org/10.5194/hess-2022-199-RC2, 2022

**(Note: All line numbers, figures, tables and sections refer to the clean copy of the manuscript)**

**Reviewer comment**: This work presents trends in annual and seasonal mean streamflow since 1950 across an expanded Australia-wide network of reference gauges. It was found that streamflow has mostly decreased, which can be thought of as a step change in the 1990s, except for some areas in the far north that saw increasing streamflow. The authors highlight the widespread interannual persistence of streamflow anomalies in Australia, and make use of statistical trend tests that account for autocorrelation. This is a valuable contribution and in my view should be published, subject to minor revisions.

*Author Response: We thank the reviewer for positive comments and finding our manuscript suitable for publication by HESS. We addressed all the comments by the reviewer as described below.*

**Reviewer comment**: There are a few unclear sentences, such as at line 383: "The main objective is to assess whether the number of locations with significant trends occur at a regional scale or not" and line 540: "It is least sensitive to outliers, and skewed distribution makes it most suitable for the analysis of streamflow data". These should be rephrased. In general, some proofreading is needed.

*Author Response: We thank the reviewer for the comment.*

*We have modified the text in line 383 (now line xx) as "The main objective is to assess whether a certain minimum number of locations with significant trends occur at a regional scale to make it field significant or not". We have also modified the text in line 540 (now line xx) as "As annual streamflow has a skewed distribution, and may have outliers, this non-parametric test, which is least sensitive to these characteristics, is well suited for change point detection".*

**Reviewer comment**: In general, some proofreading is needed.

*Author Response: As suggested by the reviewer, we have undertaken proof reading of the entire manuscript and made editorial changes as required.*

**Reviewer comment**: The terminology "linear trend" is confusing at times (e.g. Section 3.1), as the MK test is for monotonic but not necessarily linear changes (including step changes).

*Author Response: We thank the reviewer for noting this important observation. To avoid confusion, we have changed the title of Section "3.1 Linear Trend" to "3.1 Trend analyses", and replaced 'linear' to monotonic throughout the manuscript.*

**Reviewer comment**: In Section 3.3, mention that the test for regional significance is actually conservative, as it is based on a null hypothesis of independent trends across stations, when the trends within a region are actually positively correlated.

*Author Response: We thank the reviewer for noting this point. We completely agree with the reviewer and have now included this information in Section 3.3, and modified the text accordingly, Lines 380-382*

> The test for regional significance is rather conservative, as it is based on a null hypothesis of independent trends across stations, when the trends within a region are positively correlated.

**Reviewer comment**: Check units -- e.g., at line 409, should it be 1.8 mm/year per year?

*Author Response: We have incorporated this change as ".. streamflow is 5.91 GL/year per year (1.8 mm/year per year)".–We have also updated this unit throughout the manuscript.*

**Reviewer comment**: In Section 5.3, mention the possible role of $CO_2$ increase in reducing vegetation evapotranspiration rate, which could increasing streamflow in certain climatic and geomorphic settings, offsetting the increased evaporation rate due to warming -- cf. for example my 2008 HESS paper "Mapping and attribution of change in streamflow in the coterminous United States".

*Author Response: We thank the reviewer for this suggestion. We have now included this statement in Section 5.3 and cited the reference. Lines716-720*

> Increasing atmospheric $CO_2$ concentration may reduce evapotranspiration rates, which could lead to increased streamflow in certain climatic and geomorphic settings, and this may offset increased evaporation rates due to global warming (Krakauer and Fung, 2008). Recent studies covering south-east of Western Australia suggest the increase in evapotranspiration per unit of rainfall play an important role in streamflow reduction (Fowler et al., in review; Peterson et al., 2021).

**Reviewer comment**: In Section 5.4, consider mentioning that, given the decadal persistence in streamflow regimes, it will be useful to add the available information on streamflows before 1950 to a future analysis to better separate trends from oscillations.

*Author Response: We thank the reviewer for this important comment. We have now included this statement in Section 5.4. Lines734-735*

In future analyses, streamflow data obtained before the 1950s should be considered for identifying decadal persistence, variability and trends.

**Comment on hess-2022-199**

Conrad Wasko Community Comment #1
* * *
Referee comment on "Regional significance of historical trends and step changes in Australian streamflow" by Gnanathikkam Amirthanathan et al., Hydrol. Earth Syst. Sci. Discuss., https://doi.org/10.5194/hess-2022-199-CC1, 2022
* * *
**(Note: All line numbers, figures, tables and sections refer to the clean copy of the manuscript)**

**Reviewer comment**: As someone who has been using this world leading data set, I very much welcome this contribution. I enjoyed reading this manuscript and hope my suggestions are useful to the authors.

*Author Response: We thank the reviewer for positive comments and appreciation. Yes, your suggestions will definitely enrich the paper.*

**Reviewer comment**: Line 82 & 697: "However, it was not clear how these changes relate to change in rainfall", and "Further research is required to reveal the association of historical rainfall changes with observed streamflow". I would argue there is literature that addresses potential drivers, that is rainfall and secondly soil moisture (Wasko et al., 2021; Wasko and Nathan, 2019).

*Author Response: We thank the reviewer for the suggestions. We have elaborated the appropriate sections of the manuscript (Lines 82 and 697) and will include the references.*

Lines81

> Rainfall is one potential driver of these changes.

Lines 698-700

> Research has revealed the association of historical rainfall and soil moisture changes with observed streamflow across different drainage divisions of Australia (Wasko et al., 2021; Wasko and Nathan, 2019).

**Reviewer comment**: Line 88: I agree but would note that another manuscript focussing on Australia found flood peak timing shifting alongside rainfall peak timing for frequent floods (Wasko et al.,2020).

*Author Response: We thank the reviewer for the observation. We have now included this reference.*

**Reviewer comment**: Line 92: The following manuscript may be relevant (Gu et al., 2020)

*Author Response: Thanks for suggesting additional reference. We have modified the text and added it accordingly. Lines 93-94*

> Analyses of 780 unregulated catchments (Gu et al., 2020) reveal similar geographical distribution in trends.

**Reviewer comment**: Line 166: When I followed the link and clicked on "water year" I got the following definition: "1 July to 30 June." This is different from what was used in this manuscript.

*Author Response: We thank the reviewer for noticing the wrong link. We have defined the 'water year' according to the Hydrologic Reference Stations 'Glossary' (http://www.bom.gov.au/water/hrs/glossary.shtml). We have now fixed the link.*

**Reviewer comment**: Line 195: Why was only mean/total streamflow considered when previously a range of percentiles was examined (Zhang et al., 2016)?

*Author Response: We thank the reviewer for noticing the differences. In this paper, our focus is trends (both linear and step changes) and regional significance of annual and seasonal streamflow. Other flow components and analyses are out of scope of the paper. We have elaborated the text in the scope and objective section (Lines 113-118) properly to make this clear. Lines 115-116*

> Investigation of changes in other streamflow variables or driving forces of changes in rainfall patterns on these resulting trends in streamflow is out of scope of this study.

**Reviewer comment**: Line 230: The Pettit test is biased towards finding step changes in the centre of a time series (Mallakpour and Villarini, 2016) – though clearly the results presented here correspond well to drought periods.

*Author Response: We thank the reviewer for the observation and we agree with the comment. We have mentioned this in the revised manuscript (Section 4.2 Step change) and cited the reference. Line 541-542.*

> This test is biased towards finding step changes in the centre of a time series (Mallakpour and Villarini, 2016).

**Reviewer comment**: Line 434: You mention the MK3 test was used for short term persistence, for consistency should you mention that the MK4 test was used for long term persistence?

*Author Response: yes, we have modified the text accordingly. Lines 435*

> To analyse the presence of STP and LTP, results from the MK3 and MK4 tests were examined respectively.

**Reviewer comment**: Line 468: Were the magnitude of the trends (on a site-by-site basis) presented or are they just discussed in text?

*Author Response: Thanks for this question. We have now included the magnitude of trends in Southern and Northern divisions in Section 4.1.1, with Fig. 4(a) and (b).*

**Reviewer comment**: Line 645: Does this mean non-bolded values in the table are decreasing? This could be stated here.

*Author Response: We thank the reviewer for this question about clarity. Yes, non-bolded entries in the table indicate downward trend (p<0.10). We have now modified the text accordingly to make this clear. Line 646*

> Entries in bold and non-bold indicate positive upward and negative downward trends, respectively at p < 0.10.

**Reviewer comment**: Line 715: A recent paper (Peterson et al., 2021) and preprint (https://hess.copernicus.org/preprints/hess-2022-147/) suggest increased evapotranspiration per unit of precipitation as a driver.

*Author Response: We thank the reviewer for suggesting two reference articles. We have now reviewed these articles, and included this in the text. Lines 720*

Editorial:

**Reviewer comment**: Line 102: The reference here is missing from the reference list and was published in 2019 (not 2020).

*Author Response: We apologise for the omission; we have now fixed the error.*

**Reviewer comment**: Line 13: Insert "The" –> "The main objectives…"

Line 215: There are some additional spaces in this sentence.

Line 395: Missing 'l' in global.

*Author Response: Thanks for noticing the errors in Lines 13, 215 and 395, we have now fixed them in the revised manuscript.*

**Reviewer comment**: I am not sure Figure 8 adds much and it could possibly be omitted?

*Author Response: We thank the reviewer for the suggestion. This figure is simply to provide a typical example of how this information is presented. We have included a sentence to explain this figure. Lines 512-513.*

It illustrates how the observed values are located below or above 5$^{th}$ and 95$^{th}$ percentiles respectively.

**Reference**

BoM (Bureau of Meteorology), 2022. Average annual, seasonal and monthly rainfall, Commonwealth of Australia. Accessed at: http://www.bom.gov.au/jsp/ncc/climate_averages/rainfall/index.jsp

Fowler, K., Peel, M., Saft, M., Peterson, T., Western, A., Band, L., Petheram, C., Dharmadi, S., Tan, K. S., Zhang, L., Lane, P., Kiem, A., Marshall, L., Griebel, A., Medlyn, B., Ryu, D., Bonotto, G., Wasko, C., Ukkola, A., Stephens, C., Frost, A., Weligamage, H., Saco, P., Zheng, H., Chiew, F., Daly, E., Walker, G., Vervoort, R. W., Hughes, J., Trotter, L., Neal, B., Cartwright, I., and Nathan, R.: Explaining changes in rainfall-runoff relationships during and after Australia's Millennium Drought: a community perspective, Hydrol. Earth Syst. Sci. Discuss. [preprint], https://doi.org/10.5194/hess-2022-147, in review, 2022.

Gu, X., Zhang, Q., Li, J., Liu, J., Xu, C.Y., Sun, P., 2020. The changing nature and projection of floods across Australia. J. Hydrol. 584, 124703. https://doi.org/10.1016/j.jhydrol.2020.124703.

Mallakpour, I., Villarini, G., 2016. A simulation study to examine the sensitivity of the Pettitt test to detect abrupt changes in mean. Hydrol. Sci. J. 61, 245–254. https://doi.org/10.1080/02626667.2015.1008482.

Peterson, T.J., Saft, M., Peel, M.C., John, A., 2021. Watersheds may not recover from drought. Science 372, 745–749. https://doi.org/10.1126/science.abd5085.

Sagarika, S., Kalra, A. and Ahmad, S.: Evaluating the effect of persistence on long-term trends and analyzing step changes in streamflows of the continental United States, J. Hydrol., doi:10.1016/j.jhydrol.2014.05.002, 2014.

Wasko, C., Nathan, R., 2019. Influence of changes in rainfall and soil moisture on trends in flooding. J. Hydrol. 575, 432–441. https://doi.org/10.1016/j.jhydrol.2019.05.054.

Wasko, C., Nathan, R., Peel, M.C., 2020. Changes in Antecedent Soil Moisture Modulate Flood Seasonality in a Changing Climate. Water Resour. Res. 56, e2019WR026300. https://doi.org/10.1029/2019WR026300.

Wasko, C., Shao, Y., Vogel, E., Wilson, L., Wang, Q.J., Frost, A., Donnelly, C., 2021. Understanding trends in hydrologic extremes across Australia. J. Hydrol. 593, 125877. https://doi.org/10.1016/j.jhydrol.2020.125877.

Zhang, X. S., Amirthanathan, G. E., Bari, M. A., Laugesen, R. M., Shin, D., Kent, D. M., MacDonald, A. M., Turner, M. E. and Tuteja, N. K.: How streamflow has changed across Australia since the 1950s: Evidence from the network of hydrologic reference stations, Hydrol. Earth Syst. Sci., doi:10.5194/hess-20-3947-2016, 2016.

---

## Author Response (AR2)

**Comments to the author**:

Dear authors,

Thank you for your revised manuscript. I am pleased to accept your paper for publication subject to a minor correction.

L383-384: of locations with significant trends occur at a regional scale to make it field significant or not. It is unclear what you mean by 'field significant'? Do you mean regionally significant?

Sincerely,

Yi He, HESS Editor

**Author Response:**

Dear Editor,

Thanks for you question. We have updated the text 'field significant' to regionally significant in L383-384 and a few other places in the manuscript. Please see the track change version for details.

Kind Regards

Authors